# How to Hedge an Option Against an Adversary: Black-Scholes Pricing is Minimax Optimal

**Jacob Abernethy**
University of Michigan
jabernet@umich.edu

**Peter L. Bartlett**
University of California at Berkeley
and Queensland University of Technology
bartlett@cs.berkeley.edu

**Rafael M. Frongillo**
Microsoft Research
raf@cs.berkeley.edu

**Andre Wibisono**
University of California at Berkeley
wibisono@cs.berkeley.edu

## Abstract

We consider a popular problem in finance, option pricing, through the lens of an online learning game between Nature and an Investor. In the Black-Scholes option pricing model from 1973, the Investor can continuously hedge the risk of an option by trading the underlying asset, assuming that the asset's price fluctuates according to Geometric Brownian Motion (GBM). We consider a worst-case model, in which Nature chooses a sequence of price fluctuations under a cumulative quadratic volatility constraint, and the Investor can make a sequence of hedging decisions. Our main result is to show that the value of our proposed game, which is the "regret" of hedging strategy, converges to the Black-Scholes option price. We use significantly weaker assumptions than previous work—for instance, we allow large jumps in the asset price—and show that the Black-Scholes hedging strategy is near-optimal for the Investor even in this non-stochastic framework.

## 1 Introduction

An *option* is a financial contract that allows the purchase or sale of a given asset, such as a stock, bond, or commodity, for a predetermined price on a predetermined date. The contract is named as such because the transaction in question is optional for the purchaser of the contract. Options are bought and sold for any number of reasons, but in particular they allow firms and individuals with risk exposure to hedge against potential price fluctuations. Airlines, for example, have heavy fuel costs and hence are frequent buyers of oil options.

What ought we pay for the privilege of purchasing an asset at a fixed price on a future expiration date? The difficulty with this question, of course, is that while we know the asset's previous prices, we are uncertain as to its future price. In a seminal paper from 1973, Fischer Black and Myron Scholes introduced what is now known as the Black-Scholes Option Pricing Model, which led to a boom in options trading as well as a huge literature on the problem of derivative pricing [2]. Black and Scholes had a key insight that a firm which had sold/purchased an option could "hedge" against the future cost/return of the option by buying and selling the underlying asset as its price fluctuates. Their model is based on stochastic calculus and requires a critical assumption that the asset's price behaves according to a *Geometric Brownian Motion* (GBM) with known drift and volatility.

The GBM assumption in particular implies that (almost surely) an asset's price fluctuates continuously. The Black-Scholes model additionally requires that the firm be able to buy and sell continuously until the option's expiration date. Neither of these properties are true in practice: the stock market is only open eight hours per day, and stock prices are known to make significant jumps even

during regular trading. These and other empirical observations have led to much criticism of the Black-Scholes model.

An alternative model for option pricing was considered[1] by DeMarzo et al. [3], who posed the question: "Can we construct hedging strategies that are robust to *adversarially chosen* price fluctuations?" Essentially, the authors asked if we may consider hedging through the lens of *regret minimization in online learning*, an area that has proved fruitful, especially for obtaining guarantees robust to worst-case conditions. Within this minimax option pricing framework, DeMarzo et al. provided a particular algorithm resembling the Weighted Majority and Hedge algorithms [5, 6] with a nice bound.

Recently, Abernethy et al. [1] took the minimax option pricing framework a step further, analyzing the zero-sum game being played between an Investor, who is attempting to replicate the option payoff, and Nature, who is sequentially setting the price changes of the underlying asset. The Investor's goal is to "hedge" the payoff of the option as the price fluctuates, whereas Nature attempts to foil the Investor by choosing a challenging sequence of price fluctuations. The *value* of this game can be interpreted as the "minimax option price," since it is what the Investor should pay for the option against an adversarially chosen price path. The main result of Abernethy et al. was to show that the game value approaches the Black-Scholes option price as the Investor's trading frequency increases. Put another way, the minimax price tends to the option price under the GBM assumption. This lends significant further credibility to the Black-Scholes model, as it suggests that the GBM assumption may already be a "worst-case model" in a certain sense.

The previous result, while useful and informative, left two significant drawbacks. First, their techniques used minimax duality to compute the value of the game, but no particular hedging algorithm for the Investor is given. This is in contrast to the Black-Scholes framework (as well as to the De-Marzo et al.'s result [3]) in which a hedging strategy is given explicitly. Second, the result depended on a strong constraint on Nature's choice of price path: the multiplicative price variance is uniformly constrained, which forbids price jumps and other large fluctuations.

In this paper, we resolve these two drawbacks. We consider the problem of minimax option pricing with much weaker constraints: we restrict the *sum* over the length of the game of the squared price fluctuations to be no more than a constant $c$, and we allow arbitrary price jumps, up to a bound $\zeta$. We show that the minimax option price is exactly the Black-Scholes price of the option, up to an additive term of $O(c\zeta^{1/4})$. Furthermore, we give an explicit hedging strategy: this upper bound is achieved when the Investor's strategy is essentially a version of the Black-Scholes hedging algorithm.

## 2 The Black-Scholes Formula

Let us now briefly review the Black-Scholes pricing formula and hedging strategy. The derivation requires some knowledge of continuous random walks and stochastic calculus—Brownian motion, Itô's Lemma, a second-order partial differential equation—and we shall only give a cursory treatment of the material. For further development we recommend a standard book on stochastic calculus, e.g. [8]. Let us imagine we have an underlying asset $A$ whose price is fluctuating. We let $W(t)$ be a *Brownian motion*, also known as a *Weiner process*, with zero drift and unit variance; in particular, $W(0) = 0$ and $W(t) \sim N(0, t)$ for $t > 0$. We shall imagine that $A$'s price path $G(t)$ is described by a *geometric Brownian motion* with drift $\mu$ and volatility $\sigma$, which we can describe via the definition of a Brownian motion: $G(t) \stackrel{d}{=} \exp\{(\mu - \frac{1}{2}\sigma^2)t + \sigma W(t)\}$.

If an Investor purchases a *European call* option on some asset $A$ (say, MSFT stock) with a *strike price* of $K > 0$ that matures at time $T$, then the Investor has the right to buy a share of $A$ at price $K$ at time $T$. Of course, if the market price of $A$ at $T$ is $G(T)$, then the Investor will only "exercise" the option if $G(T) > K$, since the Investor has no benefit of purchasing the asset at a price higher than the market price. Hence, the payoff of a European call option has a profit function of the form $\max\{0, G(T) - K\}$. Throughout the paper we shall use $g_{EC}(x) := \max\{0, x - K\}$ to refer to the payout of the European call when the price of asset $A$ at time $T$ is $x$ (the parameter $K$ is implicit).

We assume the current time is $t$. The Black-Scholes derivation begins with a guess: assume that the "value" of the European call option can be described by a smooth function $\mathcal{V}(G(t), t)$, depending only on the current price of the asset $G(t)$ and the time to expiration $T - t$. We can immediately define a boundary condition on $\mathcal{V}$, since at the expiration time $T$ the value of the option is $\mathcal{V}(G(T), 0) = g_{\mathrm{EC}}(G(T))$.

So how do we arrive at a value for the option at another time point $t$? We assume the Investor has a *hedging strategy*, $\Delta(x, t)$ that determines the amount to invest when the current price is $x$ and the time is $t$. Notice that if the asset's current price is $G(t)$ and the Investor purchases $\Delta(G(t), t)$ dollars of asset $A$ at $t$, then the incremental amount of money made in an infinitesimal amount of time is $\Delta(G(t), t) \, dG/G(t)$, since $dG/G(t)$ is the instantaneous multiplicative price change at time $t$. Of course, if the earnings of the Investor are guaranteed to exactly cancel out the infinitesimal change in the value of the option $d\mathcal{V}(G(t), t)$, then the Investor is totally hedged with respect to the option payout for any sample of $G$ for the remaining time to expiration. In other words, we hope to achieve $d\mathcal{V}(G, t) = \Delta(G, t) \, dG/G$. However, by Itô's Lemma [8] we have the following useful identity:

$$d\mathcal{V}(G, t) = \frac{\partial \mathcal{V}}{\partial x} dG + \frac{\partial \mathcal{V}}{\partial t} dt + \frac{1}{2} \sigma^2 G^2 \frac{\partial^2 \mathcal{V}}{\partial x^2} dt. \tag{1}$$

Black and Scholes proposed a generic hedging strategy, that the investor should invest

$$\Delta(x, t) = x \frac{\partial \mathcal{V}}{\partial x} \tag{2}$$

dollars in the asset $A$ when the price of $A$ is $x$ at time $t$. As mentioned, the goal of the Investor is to hedge out risk so that it is always the case that $d\mathcal{V}(G, t) = \Delta(G, t) \, dG/G$. Combining this goal with Equations (1) and (2), we have

$$\frac{\partial \mathcal{V}}{\partial t} + \frac{1}{2} \sigma^2 x^2 \frac{\partial^2 \mathcal{V}}{\partial x^2} = 0. \tag{3}$$

Notice the latter is an entirely non-stochastic PDE, and indeed it can be solved explicitly:

$$\mathcal{V}(x, t) = \mathbb{E}_Y[g_{\mathrm{EC}}(x \cdot \exp(Y))] \quad \text{where} \quad Y \sim \mathcal{N}(-\tfrac{1}{2}\sigma^2(T - t), \sigma^2(T - t)) \tag{4}$$

**Remark:** While we have described the derivation for the European call option, with payoff function $g_{\mathrm{EC}}$, the analysis above does not rely on this specific choice of $g$. We refer the reader to a standard text on asset pricing for more on this [8].

## 3   The Minimax Hedging Game

We now describe a sequential decision protocol in which an Investor makes a sequence of trading decisions on some underlying asset, with the goal of hedging away the risk of some option (or other financial derivative) whose payout depends on the final price of the asset at the expiration time $T$. We assume the Investor is allowed to make a trading decision at each of $n$ time periods, and before making this trade the investor observes how the price of the asset has changed since the previous period. Without loss of generality, we can assume that the current time is $0$ and the trading periods occur at $\{T/n, \, 2T/n, \ldots, 1\}$, although this will not be necessary for our analysis.

The protocol is as follows.
1: Initial price of asset is $S = S_0$.
2: **for** $i = 1, 2, \ldots, n$ **do**
3:     Investor hedges, invests $\Delta_i \in \mathbb{R}$ dollars in asset.
4:     Nature selects a price fluctuation $r_i$ and updates price $S \leftarrow S(1 + r_i)$.
5:     Investor receives (potentially negative) profit of $\Delta_i r_i$.
6: **end for**
7: Investor is charged the cost of the option, $g(S) = g\left(S_0 \cdot \prod_{i=1}^{n}(1 + r_i)\right)$.

Stepping back for a moment, we see that the Investor is essentially trying to minimize the following objective:

$$g\left(S_0 \cdot \prod_{i=1}^{n}(1 + r_i)\right) - \sum_{i=1}^{n} \Delta_i r_i.$$

We can interpret the above expression as a form of *regret*: the Investor chose to execute a trading strategy, earning him $\sum_{i=1}^{n} \Delta_i r_i$, but in hindsight might have rather purchased the option instead, with a payout of $g\left(S_0 \cdot \prod_{i=1}^{n}(1 + r_i)\right)$. What is the best hedging strategy the Investor can execute to minimize the difference between the option payoff and the gains/losses from hedging? Indeed, how much regret may be suffered against a worst-case sequence of price fluctuations?

**Constraining Nature.** The cost of playing the above sequential game is clearly going to depend on how much we expect the price to fluctuate. In the original Black-Scholes formulation, the price volatility $\sigma$ is a major parameter in the pricing function. In the work of Abernethy et al., a key assumption was that Nature may choose any $r_1, \ldots, r_n$ with the constraint that $\mathbb{E}[r_i^2 \mid r_1, \ldots, r_{i-1}] = O(1/n)$.[2] Roughly, this constraint means that in any $\epsilon$-sized time interval, the price fluctuation variance shall be no more than $\epsilon$. This constraint, however, does not allow for large price jumps during trading. In the present work, we impose a much weaker set of constraints, described as follows:[3]

- **TotVarConstraint**: The total price fluctuation is bounded by a constant $c$: $\sum_{i=1}^{n} r_i^2 \leq c$.
- **JumpConstraint**: Every price jump $|r_i|$ is no more than $\zeta$, for some $\zeta > 0$ (which may depend on $n$).

The first constraint above says that Nature is bounded by how much, in total, the asset's price path can fluctuate. The latter says that at no given time can the asset's price jump more than a given value. It is worth noting that if $c \geq n\zeta^2$ then **TotVarConstraint** is superfluous, whereas **JumpConstraint** becomes superfluous if $c < \zeta^2$.

**The Minimax Option Price** We are now in a position to define the *value* of the sequential option pricing game using a minimax formulation. That is, we shall ask how much the Investor loses when making optimal trading decisions against worst-case price fluctuations chosen by Nature. Let $V_\zeta^{(n)}(S; c, m)$ be the value of the game, measured by the investor's *loss*, when the asset's current price is $S \geq 0$, the **TotVarConstraint** is $c \geq 0$, the **JumpConstraint** is $\zeta > 0$, the total number of trading rounds are $n \in \mathbb{N}$, and there are $0 \leq m \leq n$ rounds remaining. We define recursively:

$$V_\zeta^{(n)}(S; c, m) = \inf_{\Delta \in \mathbb{R}} \sup_{r \,:\, |r| \leq \min\{\zeta, \sqrt{c}\}} -\Delta r + V_\zeta^{(n)}((1+r)S; \; c - r^2, m - 1), \tag{5}$$

with the base case $V_\zeta^{(n)}(S; c, 0) = g(S)$. Notice that the constraint under the supremum enforces both **TotVarConstraint** and **JumpConstraint**. For simplicity, we will write $V_\zeta^{(n)}(S; c) := V_\zeta^{(n)}(S; c, n)$. This is the value of the game that we are interested in analyzing.

Towards establishing an upper bound on the value (5), we shall discuss the question of how to choose the hedge parameter $\Delta$ on each round. We can refer to a "hedging strategy" in this game as a function of the tuple $(S, c, m, n, \zeta, g(\cdot))$ that returns hedge position. In our upper bound, in fact we need only consider hedging strategies $\Delta(S, c)$ that depend on $S$ and $c$; there certainly will be a dependence on $g(\cdot)$ as well but we leave this implicit.

## 4 Asymptotic Results

The central focus of the present paper is the following question: "*For fixed c and S, what is the asymptotic behavior of the value $V_\zeta^{(n)}(S; c)$?*" and "*Is there a natural hedging strategy $\Delta(S, c)$ that (roughly) achieves this value?*" In other words, what is the minimax value of the option, as well as the optimal hedge, when we fix the variance budget $c$ and the asset's current price $S$, but let the number of rounds tend to $\infty$? We now give answers to these questions, and devote the remainder of the paper to developing the results in detail.

We consider payoff functions $g \colon \mathbb{R}_0 \to \mathbb{R}_0$ satisfying three constraints:

1. $g$ is convex.

2. $g$ is $L$-Lipschitz, i.e. $|g(x) - g(y)| \leq L|x - y|$.

3. $g$ is *eventually linear*, i.e. there exists $K > 0$ such that $g(x)$ is a linear function for all $x \geq K$; in this case we also say $g$ is $K$-*linear*.

We believe the first two conditions are strictly necessary to achieve the desired results. The $K$-linearity may not be necessary but makes our analysis possible. We note that the constraints above encompass the standard European call and put options.

Henceforth we shall let $G$ be a *zero-drift* GBM with unit volatility. In particular, we have that $\log G(t) \sim \mathcal{N}(-\frac{1}{2}t, t)$. For $S, c \geq 0$, define the function

$$U(S, c) = \mathbb{E}_G[g(S \cdot G(c))],$$

and observe that $U(S, 0) = g(S)$. Our goal will be to show that $U$ is asymptotically the minimax price of the option. Most importantly, this function $U(S, c)$ is *identical* to $\mathcal{V}(S, \frac{1}{\sigma^2}(T - c))$, the Black-Scholes value of the option in (4) when the GBM volatility parameter is $\sigma$ in the Black-Scholes analysis. In particular, analogous to to (3), $U(S, c)$ satisfies a differential equation:

$$\frac{1}{2}S^2\frac{\partial^2 U}{\partial S^2} - \frac{\partial U}{\partial c} = 0. \tag{6}$$

The following is our main result of this paper.

**Theorem 1.** *Let $S > 0$ be the initial asset price and let $c > 0$ be the variance budget. Assume we have a sequence $\{\zeta_n\}$ with $\lim_{n\to\infty} \zeta_n = 0$ and $\liminf_{n\to\infty} n\zeta_n^2 > c$. Then*

$$\lim_{n\to\infty} V_{\zeta_n}^{(n)}(S; c) = U(S, c).$$

This statement tells us that the minimax price of an option, when Nature has a total fluctuation budget of $c$, approaches the Black-Scholes price of the option when the time to expiration is $c$. This is particularly surprising since our minimax pricing framework made no assumptions as to the stochastic process generating the price path. This is the same conclusion as in [1], but we obtained our result with a significantly weaker assumption. Unlike [1], however, we do not show that the adversary's minimax optimal stochastic price path necessarily converges to a GBM. The convergence of Nature's price path to GBM in [1] was made possible by the uniform per-round variance constraint.

The previous theorem is the result of two main technical contributions. First, we prove a lower bound on the limiting value of $V_{\zeta_n}^{(n)}(S; c)$ by exhibiting a simple randomized strategy for Nature in the form of a stochastic price path, and appealing to the Lindeberg-Feller central limit theorem. Second, we prove an $O(c\zeta^{1/4})$ upper bound on the deviation between $V_{\zeta}^{(n)}(S; c)$ and $U(S, c)$. The upper bound is obtained by providing an explicit strategy for the Investor:

$$\Delta(S, c) = S\frac{\partial U(S, c)}{\partial S}$$

and carefully bounding the difference between the output using this strategy and the game value. In the course of doing so, because we are invoking Taylor's remainder theorem, we need to bound the first few derivatives of $U(S, c)$. Bounding these derivatives turns out to be the crux of the analysis; in particular, it uses the full force of the assumptions on $g$, including that $g$ is eventually linear, to avoid the pathological cases when the derivative of $g$ converges to its limiting value very slowly.

## 5 Lower Bound

In this section we prove that $U(S, c)$ is a lower bound to the game value $V_{\zeta_n}^{(n)}(S; c)$. We note that the result in this section does not use the assumptions in Theorem 1 that $\zeta_n \to 0$, nor that $g$ is convex and eventually linear. In particular, the following result also applies when the sequence $(\zeta_n)$ is a constant $\zeta > 0$.

**Theorem 2.** *Let $g\colon \mathbb{R}_0 \to \mathbb{R}_0$ be an L-Lipschitz function, and let $\{\zeta_n\}$ be a sequence of positive numbers with $\liminf_{n\to\infty} n\zeta_n^2 > c$. Then for every $S, c > 0$,*

$$\liminf_{n\to\infty} V_{\zeta_n}^{(n)}(S; c) \geq U(S, c).$$

The proof of Theorem 2 is based on correctly "guessing" a randomized strategy for Nature. For each $n \in \mathbb{N}$, define the random variables $R_{1,n}, \ldots, R_{n,n} \sim \text{Uniform}\{\pm\sqrt{c/n}\}$ i.i.d. Note that $(R_{i,n})_{i=1}^n$ satisfies **TotVarConstraint** by construction. Moreover, the assumption $\liminf_{n\to\infty} n\zeta_n^2 > c$ implies $\zeta_n > \sqrt{c/n}$ for all sufficiently large $n$, so eventually $(R_{i,n})$ also satisfies **JumpConstraint**. We have the following convergence result for $(R_{i,n})$, which we prove in Appendix A.

**Lemma 3.** *Under the same setting as in Theorem 2, we have the convergence in distribution*

$$\prod_{i=1}^n (1 + R_{i,n}) \xrightarrow{d} G(c) \quad as \quad n \to \infty.$$

*Moreover, we also have the convergence in expectation*

$$\lim_{n\to\infty} \mathbb{E}\left[ g\left( S \cdot \prod_{i=1}^n (1 + R_{i,n}) \right) \right] = U(S, c). \tag{7}$$

With the help of Lemma 3, we are now ready to prove Theorem 2.

**Proof of Theorem 2.** Let $n$ be sufficiently large such that $n\zeta_n^2 > c$. Let $R_{i,n} \sim \text{Uniform}\{\pm\sqrt{c/n}\}$ i.i.d., for $1 \leq i \leq n$. As noted above, $(R_{i,n})$ satisfies both **TotVarConstraint** and **JumpConstraint**. Then we have

$$
\begin{aligned}
V_{\zeta_n}^{(n)}(S; c) &= \inf_{\Delta_1} \sup_{r_1} \cdots \inf_{\Delta_n} \sup_{r_n} \; g\left( S \cdot \prod_{i=1}^n (1 + r_i) \right) - \sum_{i=1}^n \Delta_i r_i \\
&\geq \inf_{\Delta_1} \cdots \inf_{\Delta_n} \; \mathbb{E}\left[ g\left( S \cdot \prod_{i=1}^n (1 + R_{i,n}) \right) - \sum_{i=1}^n \Delta_i R_{i,n} \right] \\
&= \mathbb{E}\left[ g\left( S \cdot \prod_{i=1}^n (1 + R_{i,n}) \right) \right].
\end{aligned}
$$

The first line follows from unrolling the recursion in the definition (5); the second line from replacing the supremum over $(r_i)$ with expectation over $(R_{i,n})$; and the third line from $\mathbb{E}[R_{i,n}] = 0$. Taking limit on both sides and using (7) from Lemma 3 give us the desired conclusion. $\qquad\square$

## 6   Upper Bound

In this section we prove that $U(S, c)$ is an upper bound to the limit of $V_\zeta^{(n)}(S; c)$.

**Theorem 4.** *Let $g\colon \mathbb{R}_0 \to \mathbb{R}_0$ be a convex, L-Lipschitz, K-linear function. Let $0 < \zeta \leq 1/16$. Then for any $S, c > 0$ and $n \in \mathbb{N}$, we have*

$$V_\zeta^{(n)}(S; c) \leq U(S, c) + \left( 18c + \frac{8}{\sqrt{2\pi}} \right) LK\, \zeta^{1/4}.$$

We remark that the right-hand side of the above bound does not depend on the number of trading periods $n$. The key parameter is $\zeta$, which determines the size of the largest price jump of the stock. However, we expect that as the trading frequency increases, the size of the largest price jump will shrink. Plugging a sequence $\{\zeta_n\}$ in place of $\zeta$ in Theorem 4 gives us the following corollary.

**Corollary 1.** *Let $g\colon \mathbb{R}_0 \to \mathbb{R}_0$ be a convex, L-Lipschitz, K-linear function. Let $\{\zeta_n\}$ be a sequence of positive numbers with $\zeta_n \to 0$. Then for $S, c > 0$,*

$$\limsup_{n\to\infty} V_{\zeta_n}^{(n)}(S; c) \leq U(S, c).$$

Note that the above upper bound relies on the convexity of $g$, for if $g$ were concave, then we would have the reverse conclusion:

$$V_\zeta^{(n)}(S; c) \geq g(S) = g(S \cdot \mathbb{E}[G(c)]) \geq \mathbb{E}[g(S \cdot G(c))] = U(S, c).$$

Here the first inequality follows from setting all $r = 0$ in (5), and the second is by Jensen's inequality.

## 6.1 Intuition

For brevity, we write the partial derivatives $U_c(S, c) = \partial U(S, c)/\partial c$, $U_S(S, c) = \partial U(S, c)/\partial S$, and $U_{S^2}(S, c) = \partial^2 U(S, c)/\partial S^2$. The proof of Theorem 4 proceeds by providing a "guess" for the Investor's action, which is a modification of the original Black-Scholes hedging strategy. Specifically, when the current price is $S$ and the remaining budget is $c$, then the Investor invests

$$\Delta(S, c) := SU_S(S, c).$$

We now illustrate how this strategy gives rise to a bound on $V_\zeta^{(n)}(S; c)$ as stated in Theorem 4. First suppose for some $m \geq 1$ we know that $V_\zeta^{(n)}(S; c, m-1)$ is a rough approximation to $U(S, c)$. Note that a Taylor approximation of the function $r_m \mapsto U(S + Sr_m, c - r_m^2)$ around $U(S, c)$ gives us

$$U(S + Sr_m, c - r_m^2) = U(S, c) + r_m SU_S(S, c) - r_m^2 U_c(S, c) + \frac{1}{2}r_m^2 S^2 U_{S^2}(S, c) + O(r_m^3)$$

$$= U(S, c) + r_m SU_S(S, c) + O(r_m^3),$$

where the last line follows from the Black-Scholes equation (6). Now by setting $\Delta = SU_S(S, c)$ in the definition (5), and using the assumption and the Taylor approximation above, we obtain

$$V_\zeta^{(n)}(S; c, m) = \inf_{\Delta \in \mathbb{R}} \sup_{|r_m| \leq \min\{\zeta, \sqrt{c}\}} -\Delta r_m + V_\zeta^{(n)}(S + Sr_m; c - r_m^2, m-1)$$

$$\leq \sup_{r_m} -r_m SU_S(S, c) + V_\zeta^{(n)}(S + Sr_m; c - r_m^2, m-1)$$

$$= \sup_{r_m} -r_m SU_S(S, c) + U(S + Sr_m, c - r_m^2) + \text{(approx terms)}$$

$$= U(S, c) + O(r_m^3) + \text{(approx terms)}.$$

In other words, on each round of the game we add an $O(r_m^3)$ term to the approximation error. By the time we reach $V_\zeta^{(n)}(S; c, n)$ we will have an error term that is roughly on the order of $\sum_{m=1}^{n} |r_m|^3$. Since $\sum_{m=1}^{n} r_m^2 \leq c$ and $|r_m| \leq \zeta$ by assumption, we get $\sum_{m=1}^{n} |r_m|^3 \leq \zeta c$.

The details are more intricate because the error $O(r_m^3)$ from the Taylor approximation also depends on $S$ and $c$. Trading off the dependencies of $c$ and $\zeta$ leads us to the bound stated in Theorem 4.

## 6.2 Proof (Sketch) of Theorem 4

In this section we describe an outline of the proof of Theorem 4. Throughout, we assume $g$ is a convex, $L$-Lipschitz, $K$-linear function, and $0 < \zeta \leq 1/16$. The proofs of Lemma 5 and Lemma 7 are provided in Appendix B, and Lemma 6 is proved in Appendix C.

For $S, c > 0$ and $|r| \leq \sqrt{c}$, we define the *(single-round) error term* of the Taylor approximation,

$$\epsilon_r(S, c) := U(S + Sr, c - r^2) - U(S, c) - rSU_S(S, c). \tag{8}$$

We also define a sequence $\{\alpha^{(n)}(S, c, m)\}_{m=0}^{n}$ to keep track of the cumulative errors. We define this sequence by setting $\alpha^{(n)}(S, c, 0) = 0$, and for $1 \leq m \leq n$,

$$\alpha^{(n)}(S, c, m) := \sup_{|r| \leq \min\{\zeta, \sqrt{c}\}} \epsilon_r(S, c) + \alpha^{(n)}(S + Sr, c - r^2, m-1). \tag{9}$$

For simplicity, we write $\alpha^{(n)}(S, c) \equiv \alpha^{(n)}(S, c, n)$. Then we have the following result, which formalizes the notion from the preceding section that $V_\zeta^{(n)}(S; c, m)$ is an approximation to $U(S, c)$.

**Lemma 5.** *For $S, c > 0$, $n \in \mathbb{N}$, and $0 \leq m \leq n$, we have*

$$V_\zeta^{(n)}(S; c, m) \leq U(S, c) + \alpha^{(n)}(S, c, m). \tag{10}$$

It now remains to bound $\alpha^{(n)}(S, c)$ from above. A key step in doing so is to show the following bounds on $\epsilon_r$. This is where the assumptions that $g$ be $L$-Lipschitz and $K$-linear are important.

**Lemma 6.** *For $S, c > 0$, and $|r| \leq \min\{1/16,\ \sqrt{c}/8\}$, we have*

$$\epsilon_r(S, c) \leq 16LK \left( \max\{c^{-3/2},\ c^{-1/2}\}\, |r|^3 + \max\{c^{-2},\ c^{-1/2}\}\, r^4 \right). \tag{11}$$

*Moreover, for $S > 0$, $0 < c \leq 1/4$, and $|r| \leq \sqrt{c}$, we also have*

$$\epsilon_r(S, c) \leq \frac{4LK}{\sqrt{2\pi}} \cdot \frac{r^2}{\sqrt{c}}. \tag{12}$$

Using Lemma 6, we have the following bound on $\alpha^{(n)}(S, c)$.

**Lemma 7.** *For $S, c > 0$, $n \in \mathbb{N}$, and $0 < \zeta \leq 1/16$, we have*

$$\alpha^{(n)}(S, c) \leq \left( 18c + \frac{8}{\sqrt{2\pi}} \right) LK\, \zeta^{1/4}.$$

*Proof (sketch).* By unrolling the inductive definition (9), we can write $\alpha^{(n)}(S, c)$ as the supremum of $f(r_1, \dots, r_n)$, where

$$f(r_1, \dots, r_n) = \sum_{m=1}^{n} \epsilon_{r_m} \left( S \prod_{i=1}^{m-1} (1 + r_i),\ c - \sum_{i=1}^{m-1} r_i^2 \right).$$

Let $(r_1, \dots, r_n)$ be such that $|r_m| \leq \zeta$ and $\sum_{m=1}^{n} r_m^2 \leq c$. We will show that $f(r_1, \dots, r_n) \leq (18c + 8/\sqrt{2\pi})\, LK\, \zeta^{1/4}$. Let $0 \leq n_* \leq n$ be the largest index such that $\sum_{m=1}^{n_*} r_m^2 \leq c - \sqrt{\zeta}$. We split the analysis into two parts.

1. **For $1 \leq m \leq \min\{n,\ n_* + 1\}$:** By (11) from Lemma 6 and a little calculation, we have

$$\epsilon_{r_m} \left( S \prod_{i=1}^{m-1} (1 + r_i),\ c - \sum_{i=1}^{m-1} r_i^2 \right) \leq 18LK\, \zeta^{1/4}\, r_m^2.$$

   Summing over $1 \leq m \leq \min\{n,\ n_* + 1\}$ then gives us

$$\sum_{m=1}^{\min\{n,\, n_*+1\}} \epsilon_{r_m} \left( S \prod_{i=1}^{m-1} (1+r_i),\ c - \sum_{i=1}^{m-1} r_i^2 \right) \leq 18LK\zeta^{1/4} \sum_{m=1}^{\min\{n,\, n_*+1\}} r_m^2 \leq 18LK\zeta^{1/4} c.$$

2. **For $n_* + 2 \leq m \leq n$ (if $n_* \leq n - 2$):** By (12) from Lemma 6, we have

$$\epsilon_{r_m} \left( S \prod_{i=1}^{m-1} (1 + r_i),\ c - \sum_{i=1}^{m-1} r_i^2 \right) \leq \frac{4LK}{\sqrt{2\pi}} \cdot \frac{r_m^2}{\sqrt{\sum_{i=m}^{n} r_i^2}}.$$

   Therefore,

$$\sum_{m=n_*+2}^{n} \epsilon_{r_m} \left( S \prod_{i=1}^{m-1} (1 + r_i),\ c - \sum_{i=1}^{m-1} r_i^2 \right) \leq \frac{4LK}{\sqrt{2\pi}} \sum_{m=n_*+2}^{n} \frac{r_m^2}{\sqrt{\sum_{i=m}^{n} r_i^2}} \leq \frac{8LK}{\sqrt{2\pi}} \zeta^{1/4},$$

   where the last inequality follows from Lemma 8 in Appendix B.

Combining the two cases above gives us the desired conclusion. $\qquad \square$

**Proof of Theorem 4.** Theorem 4 follows immediately from Lemma 5 and Lemma 7. $\qquad \square$

**Acknowledgments.** We gratefully acknowledge the support of the NSF through grant CCF-1115788 and of the ARC through Australian Laureate Fellowship FL110100281.

## Footnotes

[1]Although it does not have quite the same flavor, a similar approach was explored in the book of Vovk and Shafer [7].

[2]The constraint in [1] was $\mathbb{E}[r_i^2 \mid r_1, \ldots, r_{i-1}] \leq \exp(c/n) - 1$, but this is roughly equivalent.

[3]We note that Abernethy et al. [1] also assumed that the multiplicative price jumps $|r_i|$ are bounded by $\hat{\zeta}_n = \Omega(\sqrt{(\log n)/n})$; this is a stronger assumption than what we impose on $(\zeta_n)$ in Theorem 1.

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
