[Supplementary Material]

SUPPLEMENTARY MATERIAL TO
# How to Hedge an Option Against an Adversary: Black-Scholes Pricing is Minimax Optimal

**Jacob Abernethy**
University of Michigan
jabernet@umich.edu

**Peter L. Bartlett**
University of California at Berkeley
and Queensland University of Technology
bartlett@cs.berkeley.edu

**Rafael M. Frongillo**
Microsoft Research
raf@cs.berkeley.edu

**Andre Wibisono**
University of California at Berkeley
wibisono@cs.berkeley.edu

## A  Proof of Lemma 3

For each $1 \leq i \leq n$, the random variable $\log(1 + R_{i,n})$ has mean and variance, respectively,

$$\mu_n = \frac{1}{2} \log\left(1 - \frac{c}{n}\right) \quad \text{and} \quad \sigma_n^2 = \frac{1}{4} \log^2\left(\frac{\sqrt{n} + \sqrt{c}}{\sqrt{n} - \sqrt{c}}\right).$$

We now define

$$X_{i,n} := \frac{\log(1 + R_{i,n}) - \mu_n}{\sigma_n \sqrt{n}}, \tag{13}$$

so $X_{1,n}, \ldots, X_{n,n}$ are i.i.d. random variables with $\mathbb{E}[X_{i,n}] = 0$ and $\sum_{i=1}^{n} \mathbb{E}[X_{i,n}^2] = 1$. Recalling that $R_{i,n} \in \{\pm\sqrt{c/n}\}$, we see that the two possible values for $X_{i,n}$ both approach 0 as $n \to \infty$. This means for any $\epsilon > 0$ we can find a sufficiently large $n$ such that $|X_{i,n}| < \epsilon$ for all $1 \leq i \leq n$. In particular, this implies the Lindeberg condition for the triangular array $(X_{i,n}, 1 \leq i \leq n)$: for all $\epsilon > 0$,

$$\lim_{n \to \infty} \sum_{i=1}^{n} \mathbb{E}[X_{i,n}^2 \, \mathbf{1}\{|X_{i,n}| > \epsilon\}] = 0.$$

Thus, by the Lindeberg central limit theorem [4, Theorem 3.4.5], we have the convergence in distribution $\sum_{i=1}^{n} X_{i,n} \xrightarrow{d} Z$, where $Z \sim \mathcal{N}(0, 1)$ is a standard Gaussian random variable.

Clearly $\mu_n \to 0$ as $n \to \infty$. Furthermore, one can easily verify that by the L'Hôpital's rule,

$$\lim_{n \to \infty} \sigma_n \sqrt{n} = \sqrt{c} \quad \text{and} \quad \lim_{n \to \infty} n\mu_n = -\frac{c}{2}.$$

Therefore, from the convergence $\sum_{i=1}^{n} X_{i,n} \xrightarrow{d} Z$ and recalling the definition (13) of $X_{i,n}$, we also obtain

$$\sum_{i=1}^{n} \log(1 + R_{i,n}) = \sum_{i=1}^{n} \left(\mu_n + \sigma_n \sqrt{n} X_{i,n}\right) = n\mu_n + \sigma_n \sqrt{n} \sum_{i=1}^{n} X_{i,n} \xrightarrow{d} \frac{-c}{2} + \sqrt{c}Z.$$

In particular, by the continuous mapping theorem,

$$\prod_{i=1}^{n} (1 + R_{i,n}) \xrightarrow{d} \exp\left(-\frac{c}{2} + \sqrt{c}Z\right) \stackrel{d}{=} G(c).$$

We now want to show that we also have convergence in expectation when $g$ is an $L$-Lipschitz function, namely, that $\mathbb{E}[g(S \cdot \prod_{i=1}^{n}(1 + R_{i,n}))] \rightarrow \mathbb{E}[g(S \cdot G(c))]$. Without loss of generality (by replacing $g(x)$ by $\hat{g}(x) = g(S \cdot x) - g(0)$) we may assume $S = 1$ and $g(0) = 0$. For simplicity, let $S_n = \prod_{i=1}^{n}(1 + R_{i,n})$. For each $M > 0$ define the continuous bounded function $g_M(x) = \min\{g(x), M\}$. The convergence in distribution $S_n \xrightarrow{d} G(c)$ gives us

$$\lim_{n\to\infty} \mathbb{E}[g_M(S_n))] = \mathbb{E}[g_M(G(c))] \quad \text{for all } M > 0. \tag{14}$$

Since $g_M \uparrow g$ pointwise, by the monotone convergence theorem we also have

$$\lim_{M\to\infty} \mathbb{E}[g_M(G(c))] = \mathbb{E}[g(G(c))]. \tag{15}$$

Now observe that $\mathbb{E}[S_n] = 1$ and $\mathbb{E}[S_n^2] = (1 + c/n)^n \leq \exp(c)$. Since $g(0) = 0$ and $g$ is $L$-Lipschitz, we have $g(x) \leq Lx$ for all $x \geq 0$. In particular, $\mathbb{E}[g(S_n)^2] \leq L^2 \mathbb{E}[S_n^2] \leq L^2 \exp(c)$. Moreover, by Markov's inequality,

$$\mathbb{P}(g(S_n) > M) \leq \mathbb{P}\left(S_n > \frac{M}{L}\right) \leq \frac{\mathbb{E}[S_n]}{M/L} = \frac{L}{M}.$$

Therefore, for each $n$ and for all $M > 0$, by Cauchy-Schwarz inequality,

$$\begin{aligned}
\left|\mathbb{E}[g(S_n)] - \mathbb{E}[g_M(S_n)]\right| &= \mathbb{E}[(g(S_n) - M) \cdot \mathbf{1}\{g(S_n) > M\}] \\
&\leq \mathbb{E}[g(S_n) \cdot \mathbf{1}\{g(S_n) > M\}] \\
&\leq \mathbb{E}[g(S_n)^2]^{1/2}\, \mathbb{P}(g(S_n) > M)^{1/2} \\
&\leq \left(L^3 \exp(c)/M\right)^{1/2}.
\end{aligned}$$

Since the final bound does not involve $n$, this shows that $\lim_{M\to\infty} \mathbb{E}[g_M(S_n)] \rightarrow \mathbb{E}[g(S_n)]$ uniformly in $n$. This allows us to interchange the order of the limit operations below, which, together with (14) and (15), give us our desired result:

$$\lim_{n\to\infty} \mathbb{E}[g(S_n)] = \lim_{n\to\infty}\lim_{M\to\infty} \mathbb{E}[g_M(S_n)] = \lim_{M\to\infty}\lim_{n\to\infty} \mathbb{E}[g_M(S_n)] = \lim_{M\to\infty} \mathbb{E}[g_M(G(c))] = \mathbb{E}[g(G(c))].$$

This completes the proof of Lemma 3.

## B  Proofs of Lemma 5 and Lemma 7

### B.1  Proof of Lemma 5

Lemma 5 essentially follows from the definition of $\alpha^{(n)}$.

**Proof of Lemma 5.** We proceed by induction on $m$. For the base case $m = 0$, we use Jensen's inequality and the fact that $\mathbb{E}[G(c)] = 1$:

$$V_\zeta^{(n)}(S; c, 0) = g(S) = g(S \cdot \mathbb{E}[G(c)]) \leq \mathbb{E}[g(S \cdot G(c))] = U(S, c).$$

Now assume the statement (10) holds for $m - 1$. Then for $m$,

$$\begin{aligned}
V_\zeta^{(n)}(S; c, m) &= \inf_{\Delta\in\mathbb{R}} \sup_{|r|\leq\min\{\zeta,\sqrt{c}\}} -\Delta r + V_\zeta^{(n)}(S + Sr; c - r^2, m - 1) \\
&\leq \inf_{\Delta\in\mathbb{R}} \sup_{|r|\leq\min\{\zeta,\sqrt{c}\}} -\Delta r + U(S + Sr, c - r^2) + \alpha^{(n)}(S + Sr, c - r^2, m - 1) \\
&\leq \sup_{|r|\leq\min\{\zeta,\sqrt{c}\}} -rSU_S(S, c) + U(S + Sr, c - r^2) + \alpha^{(n)}(S + Sr, c - r^2, m - 1) \\
&= \sup_{|r|\leq\min\{\zeta,\sqrt{c}\}} U(S, c) + \epsilon_r(S, c) + \alpha^{(n)}(S + Sr, c - r^2, m - 1) \\
&= U(S, c) + \alpha^{(n)}(S, c, m).
\end{aligned}$$

The first line is from the definition (5); the second line is using the inductive hypothesis that (10) holds for $m-1$; the third line is from substituting the choice $\Delta = SU_S(S, c)$; the fourth line is from the definition of $\epsilon_r$; and the last line is from the definition of $\alpha^{(n)}(S, c, m)$. $\qquad\square$

## B.2 Proof of Lemma 7

For completeness, we provide a more detailed proof of Lemma 7.

**Proof of Lemma 7.** Unrolling the inductive definition (9), we can write

$$\alpha^{(n)}(S,c) = \sup_{\substack{r_1,\ldots,r_n \\ |r_m| \le \zeta,\ \sum_{m=1}^{n} r_m^2 \le c}} f(r_1,\ldots,r_n),$$

where $f$ is the function

$$f(r_1,\ldots,r_n) = \sum_{m=1}^{n} \epsilon_{r_m}\Big( S \prod_{i=1}^{m-1}(1+r_i),\ c - \sum_{i=1}^{m-1} r_i^2 \Big).$$

Let $(r_1,\ldots,r_n)$ be such that $|r_m| \le \zeta$ and $\sum_{m=1}^{n} r_m^2 \le c$. We will show that $f(r_1,\ldots,r_n) \le (18c + 8/\sqrt{2\pi})\,LK\,\zeta^{1/4}$.

Assume for now that $\zeta \le c^2$. Let $0 \le n_* \le n$ be the largest index such that

$$\sum_{m=1}^{n_*} r_m^2 \le c - \sqrt{\zeta}.$$

We split the analysis into two parts.

**For $1 \le m \le \min\{n,\, n_* + 1\}$:** We want to apply the bound in Lemma 6, so let us verify that the conditions in Lemma 6 are satisfied. Clearly $|r_m| \le \zeta \le 1/16$. Moreover, since $c - \sum_{i=1}^{m-1} r_i^2 \ge c - \sum_{i=1}^{n_*} r_i^2 \ge \sqrt{\zeta}$ and $\zeta \le 1/16$, we also have

$$|r_m| \le \zeta \le \frac{\zeta^{1/4}}{8} \le \frac{\sqrt{c - \sum_{i=1}^{m-1} r_i^2}}{8}.$$

Therefore, by (11) from Lemma 6,

$$
\begin{aligned}
\epsilon_{r_m}\Big( S \prod_{i=1}^{m-1}(1+r_i),\ c - \sum_{i=1}^{m-1} r_i^2 \Big) &\le 16LK\,\Bigg( \max\Big\{ \Big(c - \sum_{i=1}^{m-1} r_i^2\Big)^{-3/2},\ \Big(c - \sum_{i=1}^{m-1} r_i^2\Big)^{-1/2} \Big\}\,|r_m|^3 \\
&\qquad\qquad + \max\Big\{ \Big(c - \sum_{i=1}^{m-1} r_i^2\Big)^{-2},\ \Big(c - \sum_{i=1}^{m-1} r_i^2\Big)^{-1/2} \Big\}\,r_m^4 \Bigg) \\
&\le 16LK\,\Big( \max\{\zeta^{-3/4},\ \zeta^{-1/4}\}\,|r_m|^3 + \max\{\zeta^{-1},\ \zeta^{-1/4}\}\,r_m^4 \Big) \\
&= 16LK\,\Big( \zeta^{-3/4}\,|r_m|^3 + \zeta^{-1}\,r_m^4 \Big) & \text{(since } \zeta < 1) \\
&\le 16LK\,\Big( \zeta^{1/4}\,r_m^2 + \zeta\,r_m^2 \Big) & \text{(since } |r_m| \le \zeta) \\
&\le 16LK\,\Big( \zeta^{1/4}\,r_m^2 + \zeta^{1/4}\,\frac{1}{16^{3/4}}\,r_m^2 \Big) & \text{(since } \zeta \le 1/16) \\
&= 18LK\,\zeta^{1/4}\,r_m^2.
\end{aligned}
$$

Summing over $1 \le m \le \min\{n,\, n_* + 1\}$ gives us

$$\sum_{m=1}^{\min\{n,\, n_*+1\}} \epsilon_{r_m}\Big( S \prod_{i=1}^{m-1}(1+r_i),\ c - \sum_{i=1}^{m-1} r_i^2 \Big) \le 18LK\zeta^{1/4} \sum_{m=1}^{\min\{n,\, n_*+1\}} r_m^2 \le 18LK\zeta^{1/4}c. \quad (16)$$

**For $n_* + 2 \le m \le n$, if $n_* \le n - 2$:** Without loss of generality we may assume $r_n \neq 0$, for if $r_n = 0$, then the term depending on $r_n$ does not affect $f(r_1,\ldots,r_n)$ since

$$\epsilon_{r_n}\Big( S \prod_{i=1}^{n-1}(1+r_i),\ c - \sum_{i=1}^{n-1} r_i^2 \Big) = 0,$$

so we can remove $r_n$ and only consider $n_* + 2 \leq m \leq n - 1$. From the definition of $n_*$ we see that $\sum_{m=1}^{n_*+1} r_m^2 > c - \sqrt{\zeta}$, and since $\sum_{m=1}^n r_m^2 \leq c$, this implies

$$\sum_{m=n_*+2}^{n} r_m^2 \leq c - \sum_{m=1}^{n_*+1} r_m^2 < c - (c - \sqrt{\zeta}) = \sqrt{\zeta}. \tag{17}$$

Note also that for each $n_* + 2 \leq m \leq n$,

$$0 < r_n^2 \leq \sum_{i=m}^{n} r_i^2 \leq c - \sum_{i=1}^{m-1} r_i^2 \leq c - \sum_{i=1}^{n_*+1} r_i^2 \leq \sqrt{\zeta} \leq \frac{1}{4},$$

so by (12) from Lemma 6,

$$\epsilon_{r_m}\left(S \prod_{i=1}^{m-1}(1 + r_i),\ c - \sum_{i=1}^{m-1} r_i^2\right) \leq \frac{4LK}{\sqrt{2\pi}} \cdot \frac{r_m^2}{\sqrt{c - \sum_{i=1}^{m-1} r_i^2}} \leq \frac{4LK}{\sqrt{2\pi}} \cdot \frac{r_m^2}{\sqrt{\sum_{i=m}^{n} r_i^2}}.$$

Therefore, by applying Lemma 8 below to $x_i = r_{n_*+1+i}^2$, we see that

$$\sum_{m=n_*+2}^{n} \epsilon_{r_m}\left(S \prod_{i=1}^{m-1}(1 + r_i),\ c - \sum_{i=1}^{m-1} r_i^2\right) \leq \frac{4LK}{\sqrt{2\pi}} \sum_{m=n_*+2}^{n} \frac{r_m^2}{\sqrt{\sum_{i=m}^{n} r_i^2}}$$
$$\leq \frac{8LK}{\sqrt{2\pi}}\left(\sum_{m=n_*+2}^{n} r_m^2\right)^{1/2} \leq \frac{8LK}{\sqrt{2\pi}}\zeta^{1/4}, \tag{18}$$

where the last inequality follows from (17). Combining (16) and (18) gives us the desired conclusion.

Now if $\zeta > c^2$, then the argument in the second case above (for $n_* + 2 \leq m \leq n$) still holds with $n_*$ set to be $-1$, so we still get the same conclusion. $\qquad\square$

It now remains to prove the following result, which we use at the end of the proof of Lemma 7.

**Lemma 8.** *For $x_1, \ldots, x_k \geq 0$ with $x_k > 0$, we have*

$$\sum_{i=1}^{k} \frac{x_i}{\sqrt{x_i + x_{i+1} + \cdots + x_k}} \leq 2\left(\sum_{i=1}^{k} x_i\right)^{1/2}.$$

*Proof.* Let $\mathcal{L}_k$ denote the objective function that we wish to bound,

$$\mathcal{L}_k(x_1, \ldots, x_k) = \sum_{i=1}^{k} \frac{x_i}{\sqrt{x_i + x_{i+1} + \cdots + x_k}},$$

and note that for any $t > 0$,

$$\mathcal{L}_k(tx_1, \ldots, tx_k) = \sqrt{t}\,\mathcal{L}_k(x_1, \ldots, x_k), \tag{19}$$

For each $k \in \mathbb{N}$, let $\Delta_k$ denote the unit simplex in $\mathbb{R}^k$ with $x_k > 0$,

$$\Delta_k = \left\{(x_1, \ldots, x_k) : x_1, \ldots, x_{k-1} \geq 0,\ x_k > 0,\ \sum_{i=1}^{k} x_i = 1\right\},$$

and let $\eta_k$ denote the supremum of the function $\mathcal{L}_k$ over $x \in \Delta_k$. Given $x = (x_1, \ldots, x_k) \in \Delta_k$, define $y = (y_1, \ldots, y_{k-1})$ by $y_i = x_{i+1}/(1 - x_1)$, so $y \in \Delta_{k-1}$. Then we can write

$$\mathcal{L}_k(x_1, \ldots, x_k) = \frac{x_1}{\sqrt{x_1 + \cdots + x_k}} + \mathcal{L}_{k-1}(x_2, \ldots, x_k)$$
$$= x_1 + \sqrt{1 - x_1}\,\mathcal{L}_{k-1}(y_1, \ldots, y_{k-1})$$
$$\leq x_1 + \sqrt{1 - x_1}\,\eta_{k-1},$$

where the second equality is from (19) and the last inequality is from the definition of $\eta_{k-1}$. The function $x_1 \mapsto x_1 + \sqrt{1 - x_1}\,\eta_{k-1}$ is concave and maximized at $x_1^* = 1 - \eta_{k-1}^2/4$, giving us

$$\mathcal{L}_k(x_1, \ldots, x_k) \leq x_1^* + \sqrt{1 - x_1^*}\,\eta_{k-1} = 1 - \frac{\eta_{k-1}^2}{4} + \sqrt{\frac{\eta_{k-1}^2}{4}}\,\eta_{k-1} = 1 + \frac{\eta_{k-1}^2}{4}.$$

Taking the supremum over $x \in \Delta_k$ gives us the recursion

$$\eta_k \leq 1 + \frac{\eta_{k-1}^2}{4},$$

which, along with the base case $\eta_1 = 1$, easily implies $\eta_k \leq 2$ for all $k \in \mathbb{N}$. Now given $x_1, \ldots, x_k \geq 0$ with $x_k > 0$, let $x' = (tx_1, \ldots, tx_k)$ with $t = 1/(x_1 + \cdots + x_k)$, so $x' \in \Delta_k$. Then using (19) and the bound $\eta_k \leq 2$, we get

$$\mathcal{L}_k(x_1, \ldots, x_k) = \frac{1}{\sqrt{t}}\,\mathcal{L}_k(tx_1, \ldots, tx_k) \leq \eta_k \left(\sum_{i=1}^{k} x_i\right)^{1/2} \leq 2\left(\sum_{i=1}^{k} x_i\right)^{1/2},$$

as desired. $\qquad\square$

## C  Proof of Lemma 6

In this section we provide a proof of Lemma 6. Throughout the rest of this paper, we use the following notation for the higher-order partial derivatives of $U$,

$$U_{S^a c^b}(S, c) = \frac{\partial^{a+b} U(S, c)}{\partial S^a \partial c^b}, \quad a, b \in \mathbb{N}_0.$$

We will use the following bounds on $U_{S^2}$, $U_{S^3}$, and $U_{S^4}$, which we prove in Appendix D. These bounds are where we use the crucial assumptions that the payoff function $g$ is convex, $L$-Lipschitz, and $K$-linear.

**Lemma 9.** *Let $g \colon \mathbb{R}_0 \to \mathbb{R}_0$ be a convex, $L$-Lipschitz, $K$-linear function. Then for all $S, c > 0$,*

$$|U_{S^2}(S, c)| \leq \frac{2LK}{\sqrt{2\pi}} \cdot \frac{1}{S^2 \sqrt{c}} \tag{20}$$

$$|U_{S^3}(S, c)| \leq 7LK \cdot \frac{\max\{c^{-3/2},\, c^{-1/2}\}}{S^3}, \tag{21}$$

$$|U_{S^4}(S, c)| \leq 28LK \cdot \frac{\max\{c^{-2},\, c^{-1/2}\}}{S^4}. \tag{22}$$

We will also use the following property of the function $U$.

**Lemma 10.** *The function $U(S, c)$ is convex in $S$ and non-decreasing in $c$.*

*Proof.* For each fixed $c \geq 0$ and for each realization of the random variable $G(c) > 0$, the function $S \mapsto g(S \cdot G(c))$ is convex. Therefore, $U(S, c)$ is convex in $S$, being a nonnegative linear combination of convex functions. In particular, this implies $U_{S^2}(S, c) \geq 0$. So by the Black-Scholes equation (6), we also have $U_c(S, c) = \frac{1}{2} S^2 U_{S^2}(S, c) \geq 0$. $\qquad\square$

We are now ready to prove Lemma 6. For clarity, we divide the proof into two parts: we first prove the bound (12), then prove the bound (11).

**Proof of** (12) **in Lemma 6.** Recall that $U(S, c)$ is non-decreasing in $c$ by Lemma 10. Then by the Taylor remainder theorem, we can write

$$\begin{aligned}
\epsilon_r(S, c) &= U(S + Sr, c - r^2) - U(S, c) - rSU_S(S, c) \\
&\leq U(S + Sr, c) - U(S, c) - rSU_S(S, c) \\
&= \frac{1}{2} r^2 S^2 U_{S^2}(S + S\xi, c)
\end{aligned}$$

where $\xi$ is some value between $0$ and $r$. Since $|\xi| \le |r| \le \sqrt{c} \le 1/2$, we have $(1+\xi)^2 \ge 1/4$. Moreover, from (20) in Lemma 9, we have

$$\left|(1+\xi)^2 S^2 U_{S^2}(S+S\xi,c)\right| \le \frac{2LK}{\sqrt{2\pi}} \cdot \frac{1}{\sqrt{c}}.$$

Combining the bounds above gives us

$$\epsilon_r(S,c) \le \frac{1}{2}\frac{r^2}{(1+\xi)^2}\left|(1+\xi)^2 S^2 U_{S^2}(S+S\xi,c)\right| \le \frac{4LK}{\sqrt{2\pi}} \cdot \frac{r^2}{\sqrt{c}},$$

as desired. $\qquad\square$

**Proof of** (11) **in Lemma 6.** Fix $S, c > 0$, and consider the function

$$f(r) = U(S + Sr, c - r^2), \quad |r| \le \sqrt{c}.$$

By repeatedly applying the Black-Scholes differential equation (6), we can easily verify that $f(0) = U(S,c)$, $f'(0) = SU_S(S,c)$, and

$$f''(r) = p_2(r)\, r\, S^2 U_{S^2}(S+Sr, c-r^2) \ +\ p_3(r)\,(1+r)^2 r\, S^3 U_{S^3}(S+Sr, c-r^2) \\ +\ (1+r)^4 r^2\, S^4 U_{S^4}(S+Sr, c-r^2), \tag{23}$$

where $p_2, p_3$ are the polynomials $p_2(r) = 2r^3 + 4r^2 - 3r - 6$ and $p_3(r) = 4r^2 + 4r - 2$.

Noting that we can write

$$\epsilon_r(S,c) = f(r) - f(0) - f'(0)r,$$

another application of Taylor's remainder theorem allows us to write

$$\epsilon_r(S,c) = \frac{1}{2}f''(\xi)r^2$$

for some $\xi$ lying between $0$ and $r$. It is easy to verify that we have

$$\left|\frac{p_2(\xi)}{(1+\xi)^2}\right| \le 7, \qquad \left|\frac{p_3(\xi)}{(1+\xi)}\right| \le 3 \qquad \text{for all } |\xi| \le |r| \le \frac{1}{16}.$$

Moreover, since $\xi^2 \le r^2 \le c/64$, we have $c - \xi^2 \ge \frac{63}{64}c$. Then from the bound (20) in Lemma 9, we have

$$\left|(1+\xi)^2 S^2 U_{S^2}(S+S\xi, c-\xi^2)\right| \le \frac{2LK}{\sqrt{2\pi}} \cdot \frac{1}{(c-\xi^2)^{1/2}} \le \frac{2LK}{\sqrt{2\pi}} \cdot \frac{1}{(\frac{63}{64}c)^{1/2}} \le LK\, c^{-1/2}.$$

We also get from the bound (21) in Lemma 9,

$$\left|(1+\xi)^3 S^3 U_{S^3}(S+S\xi, c-\xi^2)\right| \le 7LK\ \max\{(c-\xi^2)^{-3/2}, (c-\xi^2)^{-1/2}\} \\ \le 7LK\ \max\left\{\left(\frac{63}{64}c\right)^{-3/2}, \left(\frac{63}{64}c\right)^{-1/2}\right\} \\ \le 7LK\ \left(\frac{64}{63}\right)^{3/2} \max\{c^{-3/2}, c^{-1/2}\} \\ \le 8LK\ \max\{c^{-3/2}, c^{-1/2}\}.$$

Similarly, the bound (22) in Lemma 9 gives us

$$\left|(1+\xi)^4 S^4 U_{S^4}(S+S\xi, c-\xi^2)\right| \le 29LK\ \max\{c^{-2}, c^{-1/2}\}.$$

Applying the bounds above to (23) gives us

$$|f''(\xi)| \le \left|\frac{p_2(\xi)}{(1+\xi)^2}\right| \cdot |\xi| \cdot \left|(1+\xi)^2 S^2 U_{S^2}(S+S\xi, c-\xi^2)\right| \\ + \left|\frac{p_3(\xi)}{(1+\xi)}\right| \cdot |\xi| \cdot \left|(1+\xi)^3 S^3 U_{S^3}(S+S\xi, c-\xi^2)\right| \\ + \xi^2 \cdot \left|(1+\xi)^4 S^4 U_{S^4}(S+S\xi, c-\xi^2)\right| \\ \le 7LK\,|r|\,c^{-1/2} + 24LK\,|r|\ \max\{c^{-3/2}, c^{-1/2}\} + 29LK\,r^2\ \max\{c^{-2}, c^{-1/2}\} \\ \le 31LK\,|r|\ \max\{c^{-3/2}, c^{-1/2}\} + 29LK\,r^2\ \max\{c^{-2}, c^{-1/2}\}.$$

Therefore, we obtain

$$|\epsilon_r(S,c)| = \frac{1}{2}|f''(\xi)| \cdot r^2 \le 16LK \left(|r|^3 \max\{c^{-3/2}, c^{-1/2}\} + r^4 \max\{c^{-2}, c^{-1/2}\}\right),$$

as desired.  □

## D  Proof of the Bounds on the Derivatives (Lemma 9)

In this section we prove the bounds on the higher-order derivatives $U_{S^a}(S,c)$, $a \ge 0$. Proving the bounds in Lemma 9 is more difficult than the analysis that we have done so far, and uses the full force of the assumptions that the payoff function $g$ is convex, $L$-Lipschitz, and $K$-linear.

The outline of the proof is as follows. By writing $U(S,c)$ as a convolution, we can write its derivatives $U_{S^a}(S,c)$ as an expectation of $g(S \cdot G(c))$ modulated by certain polynomials (Appendix D.1). The $K$-linearity of $g$ allows us to approximate $g$ by the European-option payoff function $g_{EC}$ that we encountered in Section 2, so we first prove Lemma 9 for the specific case when the payoff function is $g_{EC}$ (Appendix D.2). We extend the bound on $U_{S^2}(S,c)$ to the general case by dominating the function inside the expectation by another carefully constructed function (Appendix D.3). Finally, we use the approximation of $g$ by $g_{EC}$ to prove the bounds on the higher-order derivatives $U_{S^3}$ and $U_{S^4}$ (Appendix D.4). In particular, Lemma 15 proves the bound (20), and Lemma 18 proves the bounds (21) and (22).

Throughout the rest of this appendix, $Z \sim \mathcal{N}(0,1)$ denotes a standard Gaussian random variable, and $\Phi$ and $\phi$ denote the cumulative distribution function and the probability density function, respectively, of the standard Gaussian distribution. The symbol $*$ denotes the convolution operator on $\mathbb{R}$. We also use the fact that $G(c) \overset{d}{=} \exp(-\frac{1}{2}c + \sqrt{c}Z)$. Recall that the convexity of $g$ implies differentiability almost everywhere, so we can work with its derivative $g'$, which is necessarily increasing (since $g$ is convex) and satisfies $|g'(x)| \le L$ (since $g$ is $L$-Lipschitz).

Finally, in the proofs below we use the following easy property, which we state without proof.

**Lemma 11.** *For $f: \mathbb{R} \to \mathbb{R}$, $Z \sim \mathcal{N}(0,1)$, and $c \ge 0$, we have*

$$\mathbb{E}[f(Z) \exp(\sqrt{c}Z)] = \exp\left(\frac{c}{2}\right) \mathbb{E}[f(Z + \sqrt{c})],$$

*provided all the expectations above exist.*

### D.1  Formulae for the Derivatives

In this section we show that the partial derivative $U_{S^a}(S,c)$ can be expressed as an expectation of a polynomial modulated by the payoff function $g$. We define the family of polynomials $p^{[a]}(x,y)$, $a \ge 0$, as follows:

$$
\begin{aligned}
p^{[0]}(x,y) &= 1 \\
p^{[a+1]}(x,y) &= (x-ay)\,p^{[a]}(x,y) - p_x^{[a]}(x,y) \quad \text{for } a \ge 1,
\end{aligned}
\tag{24}
$$

where $p_x^{[a]}(x,y) = \partial p^{[a]}(x,y)/\partial x$.

The following is the main result in this section; note that we only assume that $g$ is Lipschitz.

**Lemma 12.** *Let $g: \mathbb{R}_0 \to \mathbb{R}_0$ be an $L$-Lipschitz function. For $a \ge 0$ and $S, c > 0$,*

$$U_{S^a}(S,c) = \frac{1}{S^a c^{a/2}} \mathbb{E}\left[p^{[a]}(Z,\sqrt{c}) \cdot g\left(S \cdot \exp\left(-\frac{c}{2} + \sqrt{c}Z\right)\right)\right],$$

*where $Z \sim \mathcal{N}(0,1)$.*

In proving Lemma 12 we will need the following result, which allows us to differentiate the convolution.

**Lemma 13.** *Fix $c > 0$. Let $g: \mathbb{R}_0 \to \mathbb{R}_0$ be an $L$-Lipschitz function, and let $\tilde{g}(x) = g(\exp(x))$. Let $\omega: \mathbb{R} \to \mathbb{R}$ be given by $\omega(x) = p(x)\,\phi(x/\sqrt{c})$, where $p(x)$ is a polynomial in $x$ with coefficients involving $c$. Finally, let $f: \mathbb{R} \to \mathbb{R}$ be given by $f(r) = (\tilde{g} * \omega)(r)$. Then the derivative $f'(r) = df(r)/dr$ can be written as the derivative of the convolution, $f'(r) = (\tilde{g} * \omega')(r)$.*

*Proof.* Fix $r \in \mathbb{R}$. For $h \neq 0$, consider the quantity $\rho_h = \frac{1}{h}(f(r+h) - f(r))$, and note that $f'(r) = \lim_{h \to 0} \rho_h$. Recalling the definition of $f$ as a convolution and using the mean-value theorem, we can write $\rho_h$ as

$$\rho_h = \int_{-\infty}^{\infty} \tilde{g}(x) \left( \frac{\omega(r - x + h) - \omega(r - x)}{h} \right) dx = \int_{-\infty}^{\infty} g(x)\, \omega'(r - x + \xi_h)\, dx,$$

for some $\xi_h$ between $0$ and $h$. Let

$$\rho_0 := \int_{-\infty}^{\infty} \tilde{g}(x)\, \omega'(r - x)\, dx = (\tilde{g} * \omega')(r).$$

Then by another application of the mean-value theorem, we can write

$$
\begin{aligned}
\Delta_h := \rho_h - \rho_0 &= \xi_h \int_{-\infty}^{\infty} \tilde{g}(x) \left( \frac{\omega'(r - x + \xi_h) - \omega'(r - x)}{\xi_h} \right) dx \\
&= \xi_h \int_{-\infty}^{\infty} \tilde{g}(x)\, \omega''(r - x + \xi_h^{(2)})\, dx,
\end{aligned}
\tag{25}
$$

for some $\xi_h^{(2)}$ lying between $0$ and $\xi_h$. One can easily verify that the second derivative of $\omega$ is given by

$$\omega''(x) = \frac{q(x)}{c^2}\, \phi\left( \frac{x}{\sqrt{c}} \right),$$

where $q(x)$ is the polynomial $q(x) = (x^2 - c)p(x) - 2cxp'(x) + c^2 p''(x)$. Since $g$ is $L$-Lipschitz, for each $x \in \mathbb{R}$ we have

$$0 \leq \tilde{g}(x) = g(\exp(x)) \leq g(0) + |g(\exp(x)) - g(0)| \leq g(0) + L\exp(x)$$

This gives us the estimate

$$
\begin{aligned}
&\left| \int_{-\infty}^{\infty} \tilde{g}(x)\, \omega''(r - x + \xi_h^{(2)})\, dx \right| \\
&\quad \leq \frac{1}{c^2} \int_{-\infty}^{\infty} \left( g(0) + L\exp(x) \right) \cdot |q(r - x + \xi_h^{(2)})| \cdot \phi\left( \frac{r - x + \xi_h^{(2)}}{\sqrt{c}} \right) dx \\
&\quad = \frac{1}{c^{3/2}} \int_{-\infty}^{\infty} \left( g(0) + L\exp(r + \xi_h^{(2)} - \sqrt{c}y) \right) \cdot |q(\sqrt{c}y)| \cdot \phi(y)\, dy < \infty,
\end{aligned}
$$

where in the computation above we have used the substitution $y = (r - x + \xi_h^{(2)})/\sqrt{c}$. The last expression above shows that the integral is finite, since we are integrating exponential and polynomial functions against the Gaussian density. Plugging this bound to (25) and recalling that $|\xi_h| \leq |h|$, we obtain

$$|\Delta_h| \leq |h| \cdot \left| \int_{-\infty}^{\infty} \tilde{g}(x)\, \omega''(r - x + \xi_h^{(2)})\, dx \right| \to 0 \quad \text{as} \quad h \to 0.$$

Since $\Delta_h = \rho_h - \rho_0$, this implies our desired conclusion,

$$f'(r) = \lim_{h \to 0} \rho_h = \rho_0 = (\tilde{g} * \omega')(r).$$

$\square$

We are now ready to prove Lemma 12.

**Proof of Lemma 12.** We proceed by induction on $a$. The base case $a = 0$ follows from the definition of $U$. Assume the statement holds for some $a \geq 0$; we prove it also holds for $a + 1$. Our strategy is to express $U_{S^a}$ as a convolution, use Lemma 13 to differentiate the convolution, and write the result back as an expectation.

Fix $S, c > 0$ for the rest of this proof. Let $\tilde{g}(x) = g(\exp(x))$ and $\phi_c(x) = \phi(x/\sqrt{c})$. From the inductive hypothesis and the fact that $-Z \stackrel{d}{=} Z$, we have

$$
\begin{aligned}
U_{S^a}(S, c) &= \frac{1}{S^a c^{a/2}} \, \mathbb{E}\left[ p^{[a]}(-Z, \sqrt{c}) \cdot \tilde{g}\left( \log S - \frac{c}{2} - \sqrt{c}Z \right) \right] \\
&= \frac{1}{S^a c^{a/2}} \int_{-\infty}^{\infty} p^{[a]}(-x, \sqrt{c}) \cdot \tilde{g}\left( \log S - \frac{c}{2} - \sqrt{c}x \right) \cdot \phi(x)\, dx \\
&= \frac{1}{S^a c^{(a+1)/2}} \int_{-\infty}^{\infty} p^{[a]}\left( -\frac{y}{\sqrt{c}}, \sqrt{c} \right) \cdot \tilde{g}\left( \log S - \frac{c}{2} - y \right) \cdot \phi_c(y)\, dy \\
&= \frac{1}{S^a c^{(a+1)/2}} \int_{-\infty}^{\infty} \tilde{g}\left( \log S - \frac{c}{2} - y \right) \cdot \omega(y)\, dy \\
&= \frac{1}{S^a c^{(a+1)/2}} (\tilde{g} * \omega)\left( \log S - \frac{c}{2} \right),
\end{aligned}
$$

where in the computation above we have used the substitution $y = \sqrt{c}x$, and we have defined the function

$$
\omega(y) = p^{[a]}\left( -\frac{y}{\sqrt{c}}, \sqrt{c} \right) \cdot \phi\left( \frac{y}{\sqrt{c}} \right).
$$

In particular, $\omega$ has derivative

$$
\omega'(y) = -\frac{1}{\sqrt{c}} \left( p_x^{[a]}\left( -\frac{y}{\sqrt{c}}, \sqrt{c} \right) + \frac{y}{\sqrt{c}} p^{[a]}\left( -\frac{y}{\sqrt{c}}, \sqrt{c} \right) \right) \phi\left( \frac{y}{\sqrt{c}} \right).
$$

Differentiating $U_{S^a}$ with respect to $S$ and using the result of Lemma 13 give us

$$
\begin{aligned}
U_{S^{a+1}}(S, c) &= -\frac{a}{S^{a+1} c^{(a+1)/2}} (\tilde{g} * \omega)\left( \log S - \frac{c}{2} \right) + \frac{1}{S^{a+1} c^{(a+1)/2}} (\tilde{g} * \omega')\left( \log S - \frac{c}{2} \right) \\
&= \frac{1}{S^{a+1} c^{(a+1)/2}} \int_{-\infty}^{\infty} \tilde{g}\left( \log S - \frac{c}{2} - y \right) (\omega'(y) - a\omega(y))\, dy \\
&= \frac{1}{S^{a+1} c^{a/2}} \int_{-\infty}^{\infty} \tilde{g}\left( \log S - \frac{c}{2} - \sqrt{c}x \right) (\omega'(\sqrt{c}x) - a\omega(\sqrt{c}x))\, dx \\
&= \frac{1}{S^{a+1} c^{a/2}} \int_{-\infty}^{\infty} \tilde{g}\left( \log S - \frac{c}{2} - \sqrt{c}x \right) \frac{((-x - a\sqrt{c}) p^{[a]}(-x, \sqrt{c}) - p_x^{[a]}(-x, \sqrt{c}))}{\sqrt{c}} \phi(x)\, dx \\
&= \frac{1}{S^{a+1} c^{(a+1)/2}} \int_{-\infty}^{\infty} \tilde{g}\left( \log S - \frac{c}{2} - \sqrt{c}x \right) p^{[a+1]}(-x, \sqrt{c}) \phi(x)\, dx \\
&= \frac{1}{S^{a+1} c^{(a+1)/2}} \, \mathbb{E}\left[ p^{[a+1]}(-Z, \sqrt{c}) \cdot \tilde{g}\left( \log S - \frac{c}{2} - \sqrt{c}Z \right) \right] \\
&= \frac{1}{S^{a+1} c^{(a+1)/2}} \, \mathbb{E}\left[ p^{[a+1]}(Z, \sqrt{c}) \cdot g\left( S \cdot \left( -\frac{c}{2} + \sqrt{c}Z \right) \right) \right],
\end{aligned}
$$

as desired. In the computation above we have again used the substitution $x = y/\sqrt{c}$ and the fact that $-Z \stackrel{d}{=} Z$. This completes the induction step and the proof of the lemma. $\qquad\square$

As an example, the first few polynomials $p^{[a]}(x, y)$ are

$$
\begin{aligned}
p^{[0]}(x, y) &= 1 \\
p^{[1]}(x, y) &= x \\
p^{[2]}(x, y) &= x^2 - yx - 1 \\
p^{[3]}(x, y) &= x^3 - 3yx^2 + (2y^2 - 3)x + 3y,
\end{aligned}
$$

giving us the formulae

$$U(S,c) = \mathbb{E}\left[g\left(S \cdot \exp\left(-\frac{c}{2} + \sqrt{c}Z\right)\right)\right]$$

$$U_S(S,c) = \frac{1}{S\sqrt{c}} \mathbb{E}\left[Z \cdot g\left(S \cdot \exp\left(-\frac{c}{2} + \sqrt{c}Z\right)\right)\right]$$

$$U_{S^2}(S,c) = \frac{1}{S^2 c} \mathbb{E}\left[(Z^2 - \sqrt{c}Z - 1) \cdot g\left(S \cdot \exp\left(-\frac{c}{2} + \sqrt{c}Z\right)\right)\right]$$

$$U_{S^3}(S,c) = \frac{1}{S^3 c^{3/2}} \mathbb{E}\left[(Z^3 - 3\sqrt{c}Z^2 + (2c-3)Z + 3\sqrt{c}) \cdot g\left(S \cdot \exp\left(-\frac{c}{2} + \sqrt{c}Z\right)\right)\right].$$

We also have the following easy corollaries.

**Corollary 2.** *For $a \geq 1$, $\mathbb{E}[p^{[a]}(Z, \sqrt{c})] = 0$. For $a \geq 2$, we also have $\mathbb{E}[p^{[a]}(Z + \sqrt{c}, \sqrt{c})] = 0$.*

*Proof.* First assume $a \geq 1$, and take $g$ to be the constant function $g(x) = 1$. In this case $U(S,c) = 1$ and $U_{S^a}(S,c) = 0$, so by the result of Lemma 12, $\mathbb{E}[p^{[a]}(Z, \sqrt{c})] = S^a c^{a/2} U_{S^a}(S,c) = 0$. Next, assume $a \geq 2$, and take $g$ to be the linear function $g(x) = x$. In this case $U(S,c) = \mathbb{E}[S \cdot G(c)] = S$, so $U_{S^a}(S,c) = 0$. Then using the results of Lemma 11 and Lemma 12,

$$\mathbb{E}\left[p^{[a]}(Z + \sqrt{c}, \sqrt{c})\right] = \exp\left(-\frac{c}{2}\right) \mathbb{E}\left[p^{[a]}(Z, \sqrt{c}) \exp(\sqrt{c}Z)\right] = S^{a-1} c^{a/2} U_{S^a}(S,c) = 0.$$

$\square$

### D.2 Calculations for the European-Option Payoff Function

In this section, we bound the derivatives $U_{S^a}(S,c)$ for the special case when $g$ is the payoff function of the European call function, $g(x) = \max\{0, x - K\}$, where $K > 0$ is a constant. Note that the bounds on $U_{S^3}$ and $U_{S^4}$ are slightly stronger than the stated bounds (21) and (22), because in this case we are able to compute the derivatives exactly.

**Lemma 14.** *Let $g(x) = \max\{0, x - K\}$. Then for all $S, c > 0$,*

$$|U_{S^2}(S,c)| \leq \frac{K}{\sqrt{2\pi}} \cdot \frac{1}{S^2\sqrt{c}}$$

$$|U_{S^3}(S,c)| \leq \frac{K}{\sqrt{2\pi}} \cdot \frac{(2\sqrt{c}+1)}{S^3 c}$$

$$|U_{S^4}(S,c)| \leq \frac{K}{\sqrt{2\pi}} \cdot \frac{(6c + 5\sqrt{c} + 2)}{S^4 c^{3/2}}$$

*Proof.* We first compute the Black-Scholes value $U(S,c)$. Define

$$\alpha \equiv \alpha(S,c) = -\frac{1}{\sqrt{c}} \log \frac{S}{K} + \frac{\sqrt{c}}{2},$$

and observe that $S \cdot \exp(-c/2 + \sqrt{c}Z) \geq K$ if and only if $Z \geq \alpha$. Then using the result of Lemma 11, we have

$$U(S,c) = \mathbb{E}\left[\left(S \cdot \exp\left(-\frac{c}{2} + \sqrt{c}Z\right) - K\right) \cdot \mathbf{1}\{Z \geq \alpha\}\right]$$

$$= S \cdot \exp\left(-\frac{c}{2}\right) \mathbb{E}\left[\exp(\sqrt{c}Z) \cdot \mathbf{1}\{Z \geq \alpha\}\right] - K\, \mathbb{P}(Z \geq \alpha)$$

$$= S\, \mathbb{P}(Z \geq \alpha - \sqrt{c}) - K\, \mathbb{P}(Z \geq \alpha)$$

$$= S\, \Phi(-\alpha + \sqrt{c}) - K\, \Phi(-\alpha).$$

Differentiating the formula above with respect to $c$ and applying the Black-Scholes differential equation (6), we get

$$U_{S^2}(S,c) = \frac{2}{S^2} U_c(S,c) = \frac{1}{S^2 c}\left[S\alpha\phi\left(-\alpha + \sqrt{c}\right) + K(-\alpha + \sqrt{c})\phi\left(\alpha\right)\right] = \frac{K}{S^2\sqrt{c}} \phi(\alpha),$$

where the last equality follows from the relation $S\phi(-\alpha+\sqrt{c}) = K\phi(\alpha)$. In particular, we have the bound $0 \le U_{S^2}(S, c) \le K/(S^2\sqrt{2\pi c})$. A direct calculation reveals that the higher order derivatives of $U$ are given by

$$U_{S^3}(S, c) = \frac{K}{S^3 c}\left(\alpha - 2\sqrt{c}\right)\phi(\alpha) \quad \text{and} \quad U_{S^4}(S, c) = \frac{K}{S^4 c^{3/2}}\left(\alpha^2 - 5\sqrt{c}\alpha + 6c - 1\right)\phi(\alpha).$$

It is not difficult to see that we have $|\alpha\exp(-\alpha^2/2)| \le 1$ and $|\alpha^2\exp(-\alpha^2/2)| \le 1$. Applying these bounds to the formulae above gives us the desired conclusion. □

### D.3  Bounding the Second Derivative $U_{S^2}(S, c)$

We now bound the second-order derivative $U_{S^2}(S, c)$ in the general case.

**Lemma 15.** *Let $g\colon \mathbb{R}_0 \to \mathbb{R}_0$ be a convex, $L$-Lipschitz, $K$-linear function. Then for all $S, c > 0$,*

$$0 \le U_{S^2}(S, c) \le \frac{2LK}{\sqrt{2\pi}} \cdot \frac{1}{S^2\sqrt{c}}.$$

*Proof.* Recall that $U(S, c)$ is convex in $S$ (Lemma 10), so $U_{S^2}(S, c) \ge 0$. If $g$ is a linear function, say $g(x) = \gamma x$ for some $0 \le \gamma \le L$, then $U(S, c) = \mathbb{E}[\gamma S \cdot G(c)] = \gamma S$. In this case $U_{S^2}(S, c) = 0$, and we are done.

Now assume $g$ is not a linear function. Since $g$ is non-negative, $L$-Lipschitz, and $K$-linear, we can find $0 \le \gamma \le L$ such that $g'(x) = \gamma$ for $x \ge K$. Moreover, since $g$ is convex and not a linear function, we also have that $\gamma > g'(0)$. Define the function $\tilde{g}\colon \mathbb{R}_0 \to \mathbb{R}_0$ by

$$\tilde{g}(x) = \frac{g(x) - g(0) - xg'(0)}{\gamma - g'(0)}, \tag{26}$$

and note that $\tilde{g}$ is an increasing, 1-Lipschitz convex function with $\tilde{g}(0) = \tilde{g}'(0) = 0, 0 \le \tilde{g}'(x) \le 1$, and $\tilde{g}'(x) = 1$ for $x \ge K$.

Consider the quantity $V(S, c) = \mathbb{E}[\tilde{g}(S \cdot G(c))]$, and note that we can write

$$V(S, c) = \frac{\mathbb{E}\left[g(S \cdot G(c)) - g(0) - g'(0) \cdot S \cdot G(c)\right]}{\gamma - g'(0)} = \frac{U(S, c) - g(0) - g'(0) \cdot S}{\gamma - g'(0)}.$$

Taking second derivative with respect to $S$ on both sides and using the fact that $0 \le \gamma - g'(0) \le 2L$, we obtain

$$0 \le U_{S^2}(S, c) = (\gamma - g'(0)) \cdot V_{S^2}(S, c) \le 2L \cdot V_{S^2}(S, c).$$

We already know that $V_{S^2}(S, c) \ge 0$ since $\tilde{g}$ is convex, so we only need to show that

$$V_{S^2}(S, c) \le \frac{K}{\sqrt{2\pi}} \cdot \frac{1}{S^2\sqrt{c}}.$$

For $0 < S \le K$, using the formula from Lemma 12 and the result of Lemma 16 below, we obtain

$$V_{S^2}(S, c) = \frac{1}{S^2 c}\mathbb{E}\left[(Z^2 - \sqrt{c}Z - 1) \cdot \tilde{g}\left(S \cdot \exp\left(-\frac{c}{2} + \sqrt{c}Z\right)\right)\right] \le \frac{1}{S^2 c}\cdot\frac{S\sqrt{c}}{\sqrt{2\pi}} \le \frac{K}{\sqrt{2\pi}}\cdot\frac{1}{S^2\sqrt{c}},$$

and for $S \ge K$, we use the result of Lemma 17 to obtain

$$V_{S^2}(S, c) = \frac{1}{S^2 c}\mathbb{E}\left[(Z^2 - \sqrt{c}Z - 1) \cdot \tilde{g}\left(S \cdot \exp\left(-\frac{c}{2} + \sqrt{c}Z\right)\right)\right] \le \frac{1}{S^2 c}\cdot\frac{K\sqrt{c}}{\sqrt{2\pi}} = \frac{K}{\sqrt{2\pi}}\cdot\frac{1}{S^2\sqrt{c}}.$$

This completes the proof of the lemma. □

It remains to prove the following two results, which we use in the proof of Lemma 15 above with $\tilde{g}$ in place of $g$. Note that the first result below does not use the assumption that $g$ is eventually linear.

**Lemma 16.** *Let $g\colon \mathbb{R}_0 \to \mathbb{R}_0$ be an increasing, nonnegative, convex, 1-Lipschitz function. Then for all $S, c > 0$,*

$$\mathbb{E}\left[(Z^2 - \sqrt{c}Z - 1) \cdot g\left(S \cdot \exp\left(-\frac{c}{2} + \sqrt{c}Z\right)\right)\right] \le \frac{S\sqrt{c}}{\sqrt{2\pi}}.$$

*Proof.* Fix $S, c > 0$, and define the following quantities:

$$t_1 = \frac{\sqrt{c} - \sqrt{c+4}}{2} \qquad\qquad t_2 = \frac{\sqrt{c} + \sqrt{c+4}}{2}$$

$$\lambda_1 = S \cdot \exp\left(-\frac{c}{2} + \sqrt{c}\, t_1\right) \qquad\qquad \lambda_2 = S \cdot \exp\left(-\frac{c}{2} + \sqrt{c}\, t_2\right)$$

$$g_1 = g(\lambda_1) \qquad\qquad g_2 = g(\lambda_2)$$

$$t_* = \frac{1}{\sqrt{c}} \log\left(\exp(\sqrt{c}t_2) - \frac{1}{S} \cdot \exp\left(\frac{c}{2}\right) \cdot (g_2 - g_1)\right).$$

Furthermore, define the function $h \colon \mathbb{R} \to \mathbb{R}_0$ by

$$h(x) = g_1 + \left(g_2 - g_1 - \lambda_2 + S \cdot \exp\left(-\frac{c}{2} + \sqrt{c}\, x\right)\right) \cdot \mathbf{1}\{x \geq t_*\}.$$

We will show that

$$\mathbb{E}\left[(Z^2 - \sqrt{c}Z - 1) \cdot g\left(S \cdot \exp\left(-\frac{c}{2} + \sqrt{c}Z\right)\right)\right] \leq \mathbb{E}\left[(Z^2 - \sqrt{c}Z - 1) \cdot h(Z)\right], \qquad (27)$$

and furthermore, we can evaluate the latter expectation explicitly:

$$\mathbb{E}\left[(Z^2 - \sqrt{c}Z - 1) \cdot h(Z)\right] = S\sqrt{c}\, \phi(t_* - \sqrt{c}) \leq \frac{S\sqrt{c}}{\sqrt{2\pi}}.$$

We begin by noting that $t_1$ and $t_2$ are the two roots of the polynomial $x^2 - \sqrt{c}x - 1$. Since $g$ is increasing and 1-Lipschitz,

$$g_2 - g_1 = g(\lambda_2) - g(\lambda_1) \leq \lambda_2 - \lambda_1 = S \cdot \exp\left(-\frac{c}{2}\right)\left(\exp(\sqrt{c}\, t_2) - \exp(\sqrt{c}\, t_1)\right).$$

Therefore, from the definition of $t_*$, we see that

$$\exp(\sqrt{c}\, t_2) - \exp(\sqrt{c}\, t_*) = \frac{1}{S} \cdot \exp\left(\frac{c}{2}\right) \cdot (g_2 - g_1) \leq \exp(\sqrt{c}\, t_2) - \exp(\sqrt{c}\, t_1),$$

so $t_1 \leq t_* \leq t_2$. Furthermore, by construction,

$$S \cdot \exp\left(-\frac{c}{2} + \sqrt{c}\, t_*\right) = S \cdot \exp\left(-\frac{c}{2} + \sqrt{c}\, t_2\right) - (g_2 - g_1) = \lambda_2 - g_2 + g_1,$$

so $h(t_*) = g_1$. This means $h$ is a continuous convex function of $x$ (although we will not actually use this property). We will now show that pointwise,

$$\phi(x) \cdot (x^2 - \sqrt{c}x - 1) \cdot g\left(S \cdot \exp\left(-\frac{c}{2} + \sqrt{c}x\right)\right) \leq \phi(x) \cdot (x^2 - \sqrt{c}x - 1) \cdot h(x). \qquad (28)$$

We consider four cases:

- Suppose $x \leq t_1$, so $x^2 - \sqrt{c}x - 1 \geq 0$. Since $g$ is increasing and nonnegative,

$$0 \leq g\left(S \cdot \exp\left(-\frac{c}{2} + \sqrt{c}x\right)\right) \leq g\left(S \cdot \exp\left(-\frac{c}{2} + \sqrt{c}\, t_1\right)\right) = g_1 = h(x).$$

- Suppose $t_1 \leq x \leq t_*$, so $x^2 - \sqrt{c}x - 1 \leq 0$. Since $g$ is increasing,

$$g\left(S \cdot \exp\left(-\frac{c}{2} + \sqrt{c}x\right)\right) \geq g\left(S \cdot \exp\left(-\frac{c}{2} + \sqrt{c}\, t_1\right)\right) = g_1 = h(x) \geq 0.$$

- Suppose $t_* \leq x \leq t_2$, so $x^2 - \sqrt{c}x - 1 \leq 0$. Since $g$ is increasing and 1-Lipschitz,

$$g\left(S \cdot \exp\left(-\frac{c}{2} + \sqrt{c}x\right)\right)$$
$$\geq g\left(S \cdot \exp\left(-\frac{c}{2} + \sqrt{c}\, t_2\right)\right) + S \cdot \exp\left(-\frac{c}{2} + \sqrt{c}x\right) - S \cdot \exp\left(-\frac{c}{2} + \sqrt{c}\, t_2\right)$$
$$= g_2 - \lambda_2 + S \cdot \exp\left(-\frac{c}{2} + \sqrt{c}x\right) = h(x) \geq 0.$$

- Suppose $x \geq t_2$, so $x^2 - \sqrt{c}x - 1 \geq 0$. Since $g$ is increasing and 1-Lipschitz,

$$g\left(S \cdot \exp\left(-\frac{c}{2} + \sqrt{c}x\right)\right)$$
$$\leq g\left(S \cdot \exp\left(-\frac{c}{2} + \sqrt{c}\,t_2\right)\right) + S \cdot \exp\left(-\frac{c}{2} + \sqrt{c}x\right) - S \cdot \exp\left(-\frac{c}{2} + \sqrt{c}\,t_2\right)$$
$$= g_2 - \lambda_2 + S \cdot \exp\left(-\frac{c}{2} + \sqrt{c}x\right) = h(x).$$

Integrating (28) over $x \in \mathbb{R}$ gives us the desired inequality (27). Let us now evaluate the expectation on the right hand side of (27). A simple computation using the properties of $Z \sim \mathcal{N}(0,1)$ gives us

$$\mathbb{E}\left[(Z^2 - \sqrt{c}Z - 1) \cdot h(Z)\right] = g_1\,\mathbb{E}\left[(Z^2 - \sqrt{c}Z - 1)\right]$$
$$+ (g_2 - g_1 - \lambda_2) \cdot \mathbb{E}\left[(Z^2 - \sqrt{c}Z - 1) \cdot \mathbf{1}\{Z \geq t_*\}\right]$$
$$+ S \cdot \exp\left(-\frac{c}{2}\right) \cdot \mathbb{E}\left[(Z^2 - \sqrt{c}Z - 1) \exp(\sqrt{c}Z) \cdot \mathbf{1}\{Z \geq t_*\}\right]$$
$$= (g_2 - g_1 - \lambda_2) \cdot (t_* - \sqrt{c})\,\phi(t_*) + St_*\,\phi(t_* - \sqrt{c})$$
$$= -S \cdot \exp\left(-\frac{c}{2} + \sqrt{c}\,t_*\right) \cdot (t_* - \sqrt{c})\,\phi(t_*) + St_*\,\phi(t_* - \sqrt{c})$$
$$= -S(t_* - \sqrt{c})\,\phi(t_* - \sqrt{c}) + St_*\,\phi(t_* - \sqrt{c})$$
$$= S\sqrt{c}\,\phi(t_* - \sqrt{c}),$$

as desired. $\qquad\square$

The following result is similar to Lemma 16, except that this result assumes the $K$-linearity of $g$ and achieves a stronger result.

**Lemma 17.** *Let $g\colon \mathbb{R}_0 \to \mathbb{R}_0$ be an increasing, nonnegative, convex, 1-Lipschitz function with the property that $g'(x) = 1$ for $x \geq K$. Then for all $S \geq K$ and $c > 0$,*

$$\mathbb{E}\left[(Z^2 - \sqrt{c}Z - 1) \cdot g\left(S \cdot \exp\left(-\frac{c}{2} + \sqrt{c}Z\right)\right)\right] \leq \frac{K\sqrt{c}}{\sqrt{2\pi}}.$$

*Proof.* This proof is similar in nature to the proof of Lemma 16, and we omit some of the details.

**Case 1:** Suppose $S \geq K\exp(\sqrt{c(c+4)}/2)$. Recall the European-option payoff function $g_{\mathrm{EC}}(x) = \max\{0, x - K\}$ from Section 2, and note that the $K$-linearity of $g$ implies $g(x) = g(K) + g_{\mathrm{EC}}(x)$ for $x \geq K$. Using the fact that $g$ is increasing and $K$-linear, we can show that for all $x \in \mathbb{R}$ we have

$$(x^2 - \sqrt{c}x - 1)\cdot g\left(S \cdot \exp\left(-\frac{c}{2} + \sqrt{c}x\right)\right) \leq (x^2 - \sqrt{c}x - 1)\cdot\left\{g(K) + g_{\mathrm{EC}}\left(S \cdot \exp\left(-\frac{c}{2} + \sqrt{c}x\right)\right)\right\}.$$

Integrating both sides above with $Z \sim \mathcal{N}(0,1)$ in place of $x$ and using the result of Lemma 14, we obtain

$$\mathbb{E}\left[(Z^2 - \sqrt{c}Z - 1) \cdot g\left(S \cdot \exp\left(-\frac{c}{2} + \sqrt{c}Z\right)\right)\right]$$
$$\leq \mathbb{E}\left[(Z^2 - \sqrt{c}Z - 1) \cdot \left\{g(K) + g_{\mathrm{EC}}\left(S \cdot \exp\left(-\frac{c}{2} + \sqrt{c}Z\right)\right)\right\}\right]$$
$$= \mathbb{E}\left[(Z^2 - \sqrt{c}Z - 1) \cdot g_{\mathrm{EC}}\left(S \cdot \exp\left(-\frac{c}{2} + \sqrt{c}Z\right)\right)\right] \leq \frac{K\sqrt{c}}{\sqrt{2\pi}}.$$

**Case 2:** Suppose $K \leq S \leq K\exp(\sqrt{c(c+4)}/2)$. Define the following quantities:

$$t_0 = \frac{\sqrt{c}}{2} - \frac{1}{\sqrt{c}}\log\frac{S}{K - g(K) + g_1}, \qquad \lambda_1 = S \cdot \exp\left(-\frac{c}{2} + \sqrt{c}\left(\frac{\sqrt{c} - \sqrt{c+4}}{2}\right)\right),$$

and $g_1 = g(\lambda_1)$. Consider the function $h_2\colon \mathbb{R} \to \mathbb{R}_0$ given by

$$h_2(x) = g_1 + \left(g(K) - g_1 - K + S \cdot \exp\left(-\frac{c}{2} + \sqrt{c}\,x\right)\right) \cdot \mathbf{1}\{x \geq t_0\}.$$

Using the fact that $g$ is increasing, 1-Lipschitz, and $K$-linear, we can show that for all $x \in \mathbb{R}$,

$$(x^2 - \sqrt{c}x - 1) \cdot g\left(S \cdot \exp\left(-\frac{c}{2} + \sqrt{c}x\right)\right) \leq (x^2 - \sqrt{c}x - 1) \cdot h_2(x).$$

Integrating both sides above with $Z \sim \mathcal{N}(0,1)$ in place of $x$, we get

$$\mathbb{E}\left[(Z^2 - \sqrt{c}Z - 1) \cdot g\left(S \cdot \exp\left(-\frac{c}{2} + \sqrt{c}Z\right)\right)\right] \leq \mathbb{E}\left[(Z^2 - \sqrt{c}Z - 1) \cdot h_2(Z)\right].$$

Following the same calculation as in the proof of Lemma 16, we can evaluate the latter expectation to be

$$\mathbb{E}\left[(Z^2 - \sqrt{c}Z - 1) \cdot h_2(Z)\right] = (K - g(K) + g_1)\sqrt{c}\,\phi(t_0) \leq \frac{K\sqrt{c}}{\sqrt{2\pi}},$$

where the last inequality follows from the relation $0 \leq g(K) - g_1 \leq K - \lambda_1$, since $g$ is increasing and 1-Lipschitz. $\qquad\square$

### D.4 Bounding the Higher-Order Derivatives

We now turn to bounding the higher-order derivatives $U_{S^3}(S,c)$ and $U_{S^4}(S,c)$. Our strategy is to approximate the eventually linear payoff function $g$ by the European-option payoff $g_{\text{EC}}$ and applying the bounds for $g_{\text{EC}}$ developed in Lemma 14.

**Lemma 18.** *Let $g\colon \mathbb{R}_0 \to \mathbb{R}_0$ be a convex, $L$-Lipschitz, $K$-linear function. Then for all $S, c > 0$,*

$$|U_{S^3}(S,c)| \leq 7LK \cdot \frac{\max\{c^{-3/2},\, c^{-1/2}\}}{S^3},$$

$$|U_{S^4}(S,c)| \leq 28LK \cdot \frac{\max\{c^{-2},\, c^{-1/2}\}}{S^4}.$$

*Proof.* Since $g$ is $L$-Lipschitz and $K$-linear, we can find $0 \leq \gamma \leq L$ such that $g(x) = g(K) + \gamma(x - K)$ for $x \geq K$. We decompose $g$ into two parts,

$$g(x) = \gamma g_{\text{EC}}(x) + g^*(x),$$

where $g_{\text{EC}}(x) = \max\{0, x - K\}$ is the European-option payoff function, and $g^*\colon \mathbb{R}_0 \to \mathbb{R}_0$ is given by $g^*(x) = g(x)$ for $0 \leq x \leq K$, and $g^*(x) = g(K)$ otherwise.

Then the Black-Scholes value $U(S,c)$ also decomposes,

$$U(S,c) = \mathbb{E}[g(S \cdot G(c))] = \gamma\,\mathbb{E}[g_{\text{EC}}(S \cdot G(c))] + \mathbb{E}[g^*(S \cdot G(c))] \equiv \gamma\,U^{\text{EC}}(S,c) + U^*(S,c),$$

and similarly for the derivatives,

$$U_{S^a}(S,c) = \gamma\,U_{S^a}^{\text{EC}}(S,c) + U_{S^a}^*(S,c), \qquad a \geq 0. \tag{29}$$

For the function $g_{\text{EC}}$, Lemma 14 tells us that for all $S, c > 0$,

$$|U_{S^3}^{\text{EC}}(S,c)| \leq \frac{3K}{\sqrt{2\pi}} \cdot \frac{\max\{c^{1/2},\, 1\}}{S^3 c}, \qquad |U_{S^4}^{\text{EC}}(S,c)| \leq \frac{13K}{\sqrt{2\pi}} \cdot \frac{\max\{c,\, 1\}}{S^4 c^{3/2}}. \tag{30}$$

Now for the second function $g^*$, we use Lemma 12 to write

$$U_{S^a}^*(S,c) = \frac{1}{S^a c^{a/2}}\,\mathbb{E}\left[p^{[a]}(Z, \sqrt{c}) \cdot g^*\left(S \cdot \exp\left(-\frac{c}{2} + \sqrt{c}Z\right)\right)\right]. \tag{31}$$

Since $\mathbb{E}[p^{[a]}(Z, \sqrt{c})] = 0$ for $a \geq 1$ (Corollary 2), we may assume that $g(0) = 0$, so $g^*(0) = 0$ as well. Since $g$ is $L$-Lipschitz, this implies

$$\sup_{x \in \mathbb{R}} |g^*(x)| = \max_{0 \leq x \leq K} |g(x)| \leq \max_{0 \leq x \leq K} Lx = LK.$$

Therefore, by applying triangle inequality and Cauchy-Schwarz inequality to (31), we get for $a \geq 1$,

$$|U_{S^a}^*(S,c)| \leq \frac{1}{S^a c^{a/2}}\,\mathbb{E}\left[\left|p^{[a]}(Z, \sqrt{c})\right| \cdot LK\right] \leq \frac{LK}{S^a c^{a/2}}\,\mathbb{E}\left[\left(p^{[a]}(Z, \sqrt{c})\right)^2\right]^{1/2}. \tag{32}$$

For $a = 3, 4$, we use the recursion (24) to compute the polynomials $p^{[a]}(Z, \sqrt{c})$, and we evaluate the expectation $\mathbb{E}[(p^{[a]}(Z, \sqrt{c}))^2]$. Plugging in this expectation to (32) with $a = 3$ gives us

$$|U_{S^3}^*(S, c)| \leq \frac{LK}{S^3 c^{3/2}} \cdot \left(4c^2 + 18c + 6\right)^{1/2} \leq \sqrt{28} \cdot LK \cdot \frac{\max\{c, 1\}}{S^3 c^{3/2}}. \tag{33}$$

Therefore, by combining the bound above with the first inequality in (30) and using the decomposition (29), we get the first part of our lemma,

$$|U_{S^3}(S, c)| \leq \frac{3}{\sqrt{2\pi}} \cdot LK \cdot \frac{\max\{c^{1/2}, 1\}}{S^3 c} + \sqrt{28} \cdot LK \cdot \frac{\max\{c, 1\}}{S^3 c^{3/2}} \leq 7LK \cdot \frac{\max\{c, 1\}}{S^3 c^{3/2}}.$$

A similar computation with $a = 4$ yields the second part of the lemma,

$$|U_{S^4}(S, c)| \leq \frac{13}{\sqrt{2\pi}} \cdot LK \cdot \frac{\max\{c, 1\}}{S^4 c^{3/2}} + \sqrt{518} \cdot LK \cdot \frac{\max\{c^{3/2}, 1\}}{S^4 c^2} \leq 28LK \cdot \frac{\max\{c^{3/2}, 1\}}{S^3 c^2}.$$

$\square$