[Reviews · NeurIPS 2013]

Submitted by Assigned_Reviewer_7

This paper considers the problem of option pricing in finance. Using a minimax approach, the authors construct a game between Nature and an Investor, prove that the game value of this game converges to the classic Black-Scholes option price, and give an explicit hedging strategy that achieves this value.

Clarity: This is a very math-heavy paper. Unfortunately, I am not very knowledgeable in the area of stochastic calculus, so I am unable to verify the correctness of the proofs in the paper and in the 14-page supplementary material. The authors do provide a reference to a standard book in stochastic calculus, but unfortunately I do not have the time to familiarize myself with the material. This is not a critisism of the paper. In fact, the authors explain the results in words very well and outline proof steps clearly for anyone who may look to build off these results further in the future.

Quality: As mentioned above, I could not verify the correctness of the results, but rigorous proofs are provided for all of the theorems. The authors clearly state the strengths of their results (explicit strategy, weaker assumptions required) over previous work.

Originality: The Nature vs. Investor game approach is not new, which the authors make clear in the introduction. This paper is basically an extension of Abernethy et al. [1] in two ways. Firstly, the assumptions required for the main result are strictly weaker than the assumptions used by Abernethy et al. Secondly, the authors provide an explicit hedging strategy for achieving the game value, whereas such a strategy was absent from [1]. While the mechanism may not be new, the analysis appears to be sufficiently original.

Significance: Given the statement above, I would describe the contributions here as incremental as the results are not extremely game-changing compared to previous work. However, the analysis appears quite difficult and novel, and the problem does appear to be addressed in a better way than previous research.

On the other hand, I am somewhat concerned about this paper's relevance to the NIPS community. After a quick scan through the technical areas listed in the call for papers, I had a hard time placing this paper in any of the given categories (although there is a "but not limited to" clause). Given the paper is purely analytical with no empirical results, perhaps this paper is more appropriate for a mathematics / ecommerce journal or conference. On that note, I was wondering if there are any other lessons to be learned here for a practitioner (Investor) beyond the fact that we now have more evidence that Black-Scholes hedging is robust.

Typo:
- line 227: "to to"

RESPONSE TO AUTHOR FEEDBACK:

While option pricing does not seem to fit the typical NIPS audience, I did fail to make the connection with stochastic optimization and sequential decision making algorithms. These are certainly appropriate areas for NIPS.
Summary: A very math-heavy paper that improves upon previous work, is clearly explained with words, and contains many rigorous proofs. However, the main contributions appear incremental and the relevance to NIPS is questionable.
Author Feedback

Author rebuttal: We would like to thank the reviewers for their helpful and thorough reviews of our work. Please see specific responses below.

== REVIEWER 5: ==

- Regarding the two different types of constraints, and how they are presented notationally, it is true that we could have presented both the jump constraint and the budget constraint as inputs to the function V. But we chose to make the jump constraint a parameter of V described by a subscript because it's not changing through one run of the investor/hedging game. The budget constraint is something that shrinks as the asset's price fluctuates. We decided to distinguish the changing variables (price, budget) from the non-changing ones (jump constraint, total number of rounds of game) this way.

- Regarding the scale of the constraint \zeta on price jumps: First, a jump of more than \sqrt{c} would exceed the variance budget in a single trading round. Second, recall that in the standard Black-Scholes analysis the GBM assumption effectively bounds the typical fluctuation to be something like \sqrt{T * \sigma^2 / n}, where T is the time to expiration, \sigma^2 is the variance parameter, and n is the number of trading rounds remaining. In our work, one can think of T \sigma^2 as the variance budget constraint c. So, in particular, a fluctuation on the order of \sqrt{c} is much larger than a fluctuation of \sqrt{c/n} which is essentially what is assumed in the Black Scholes analysis. Our bounds show how the regret depends on a fixed value of the fluctuation constraint \zeta. For our asymptotic results, we do need the fluctuation bound \zeta to tend to 0, but it can do so at a very slow rate, such as 1/\log n, as opposed to 1/\sqrt{n}. This allows for much larger jumps than would typically be seen in a GBM.

- Indeed, it would improve the paper to highlight where the convexity of g arises; it is actually a critical part of our analysis. The details are in the appendices, most notably Appendix C and D. We will work to clarify this in the final version.

== REVIEWER 6 ==

- Regarding your point (1): Yes this is a fair criticism, we should be more careful with our language here. We note that the constraint zeta <= 1/16 in Theorem 4 is only to make the constant (18c + 8/sqrt{2\pi}) explicit. A larger vale of zeta would result in a larger constant.

- Regarding your point (2): Yes this is an important observation. We should definitely say this in the paper.


== REVIEWER 7 ==

- Regarding the point on 'Clarity': We certainly don't expect the average NIPS reader to focus heavily on the stochastic calculus, and try to understand that entire literature. We simply wanted to include enough of a sketch so the reader can at least appreciate the connection we are making. The proofs in the appendix are somewhat math-heavy, but it's mostly calculations that use standard real and convex analysis. We hope to shorten the presentation for a future version.

- Regarding the point on 'Relevance': All methods for online learning (an area that has been well-represented at NIPS) can be viewed as hedging strategies that make predictions which insure against future choices of the process generating the data. We would agree that the area of "option pricing" doesn't seem an obvious fit for the NIPS audience. But this is perhaps somewhat surprising given that the Black-Scholes model relies on concepts like stochastic optimization and sequential decision algorithms for the hedging strategy design. One of the goals of this work is to draw interest in this topic, especially by showing how tools such as "online learning" and "minimax analysis" - both very much a part of the NIPS toolkit - give new insights into this classical framework. We very much hope that this will spur new research in this direction.